



# Aircraft measurements of aerosol and trace gas chemistry in the Eastern North Atlantic

Maria A. Zawadowicz[1], Kaitlyn Suski[1,c], Jiumeng Liu[1,b], Mikhail Pekour[1], Jerome Fast[1], Fan Mei[1], Arthur Sedlacek[2], Stephen Springston[2], Yang Wang[3,a], Rahul A. Zaveri[1], Robert Wood[4], Jian Wang[3], John E. Shilling[1]

[1]Atmospheric Sciences and Global Change Division, Pacific Northwest National Laboratory, Richland, WA, 99352, USA
[2]Environmental & Climate Sciences Department, Brookhaven National Laboratory, Upton, NY 11973, USA
[3]Department of Energy, Environmental and Chemical Engineering, Washington University in St. Louis, Saint Louis, MO, 63130, USA
[4]Department of Atmospheric Science, University of Washington, Seattle, WA, 98195, USA

[a]Now at: Department of Civil, Architectural and Environmental Engineering, Missouri University of Science and Technology, Rolla, MO, 65409, USA
[b]Now at: School of Environment, Harbin Institute of Technology, Harbin, Heilongjiang, China.
[c]Now at: JUUL Labs, San Francisco, CA, 94107, USA

*Correspondence to*: John E. Shilling (john.shilling@pnnl.gov)

**Abstract.** The Aerosol and Cloud Experiment in the Eastern North Atlantic (ACE-ENA) investigated properties of aerosols and subtropical marine boundary layer (MBL) clouds. Low subtropical marine clouds can have a large effect on Earth's radiative budget, but they are poorly represented in global climate models. In order to understand their radiative effects, it is imperative to understand the composition and sources of the MBL cloud condensation nuclei (CCN). The campaign consisted of two intensive operation periods (IOP) (June-July, 2017 and January-February, 2018) during which a fully instrumented G-1 aircraft was deployed from Lajes Field on Terceira Island in the Azores, Portugal. The G-1 conducted research flights in the vicinity of the Atmospheric Radiation Measurement (ARM) Eastern North Atlantic (ENA) atmospheric observatory on Graciosa Island. An Aerodyne HR-ToF Aerosol Mass Spectrometer (AMS) and Ionicon Proton-Transfer-Reaction Mass Spectrometer (PTR-MS) were deployed aboard the aircraft, characterizing chemistry of non-refractory aerosol and trace gases, respectively. The Eastern North Atlantic region was found to be very clean, with average non-refractory aerosol mass loading of 0.6 μg m$^{-3}$ in the summer and 0.1 μg m$^{-3}$ in the winter, measured by the AMS. Average concentrations of trace reactive gases methanol and acetone were 1 - 2 ppb; benzene, toluene and isoprene were even lower, < 1 ppb. Mass fractions of sulfate, organics, ammonium and nitrate in the boundary layer were 69%, 23%, 7% and 1% and remained largely similar between seasons. The aerosol chemical composition was dominated by sulfate and highly processed organics. Particulate methanesulfonic acid (MSA), a well-known secondary biogenic marine species was detected, with an average boundary layer concentration of 0.021 μg m$^{-3}$, along with its gas-phase precursor, dimethyl sulfide (DMS). MSA accounted for no more than 1% of the sulfate and no more than 3% of the total aerosol in the boundary layer. Examination of vertical profiles of aerosol and gas chemistry during ACE-ENA reveals an interplay of local marine emissions and long-range transported aged aerosol.





A case of transport of biomass burning emissions from North American fires has been identified using back-trajectory analysis. In the summer, the non-refractory portion of the background CCN budget was heavily influenced by aerosol associated with ocean productivity, in particular sulfate formed from DMS oxidation. Episodic transport from the continents, particular of biomass burning aerosol, periodically increased CCN concentrations in the free troposphere. In the winter, with ocean productivity lower, CCN concentrations were overall much lower and dominated by remote transport. These results show that

anthropogenic emissions perturb CCN concentrations in remote regions that are sensitive to changes in CCN number and illustrate that accurate predictions of both transport and regional aerosol formation from the oceans is critical to accurately modeling clouds in these regions.

## 1 Introduction

Interactions of atmospheric aerosols with clouds and precipitation are a vital, yet uncertain, part of the climate system. Indirect

aerosol radiative forcing effects are associated with large uncertainties in global climate models (Boucher et al., 2013), and they are poorly constrained, with a factor of 5 variation across different models (Quaas et al., 2009). An effect that is especially relevant to these clouds is the first indirect, or Twomey effect (Twomey, 1974), which accounts for changes to cloud albedo resulting from perturbations in aerosol concentration, and therefore the cloud condensation nuclei (CCN) availability. Remote marine clouds are particularly susceptible to perturbations in aerosol concentrations because they are relatively optically thin

and the background aerosol concentrations in the remote MBL are low (Quaas et al., 2009, Bony and Dufresne, 2005). An effect that is especially relevant to these clouds is the first indirect, or Twomey effect (Twomey, 1974), which accounts for changes to cloud albedo resulting from perturbations in aerosol concentration, and therefore the cloud condensation nuclei (CCN) availability. Remote marine clouds are particularly susceptible to perturbations in aerosol concentrations because they are relatively optically thin and the background aerosol concentrations in the remote MBL are low (Wood, 2005, Carslaw et

al., 2013). However, by itself, the Twomey effect is insufficient in explaining all indirect effects (Wood, 2007). Factors such as aerosol suppression of precipitation can also be important (Wood, 2005, Wood et al., 2015), highlighting the need for measurement of aerosol and cloud properties in the remote MBL.

Sources of CCN in the MBL have been at the center of much research for several decades. Early studies have hypothesized that dimethyl sulfide (DMS), a volatile organic compound (VOC) produced by phytoplankton, is a major source of secondary

sulfate aerosols, which then dominate the marine CCN budget (Charlson et al., 1987). This came to be known as the CLAW hypothesis, which posits a feedback loop between ocean biochemistry, marine cloud properties, and climate (Charlson et al., 1987). More recently, this view has been debated, as primary sea salt aerosols enriched in organics have been hypothesized to be a more robust source of CCN than DMS-derived aerosols (Quinn and Bates, 2011). This includes primary organic species emitted together with sea salt upon bubble bursting, which is a mixture of fragments of marine biota, their exudates and other

simple proteins, lipids and carbohydrates (Quinn and Bates, 2011). However, it has also been shown that sea salt contributes only 30% to the CCN budget in mid-latitudes, and secondary, possibly DMS-derived, non-sea salt sulfate can dominate most



MBL CCN budgets (Quinn et al., 2017). This secondary source of CCN has been shown to be important in the ENA region (Zheng et al., 2018). Because the emissions of marine VOCs and chemistry of both primary and secondary marine organic aerosol are controlled by ocean ecosystems, they are likely impacted by climate change. Sea surface temperatures drive marine

phytoplankton diversity, and the distributions of phytoplankton species in the warming ocean are likely to shift in the near future (Righetti et al., 2019, Flombaum et al., 2013). A recent modeling study by Wang et al. (2018) found significant radiative effects from such broad community shifts. Understanding complex relationships between ocean diversity, aerosol chemistry, and MBL clouds is of critical importance in the changing climate.

Apart from local oceanic sources of aerosols, such as DMS oxidation and ejection of sea salt, the remote MBL can also be

impacted by long-range continental transport. Biomass burning plumes can be effectively transported between the continents (Brocchi et al., 2018), as the aerosols are injected into the free troposphere or the stratosphere. Secondary aerosols of continental origins, such as SOA produced by oxidation of VOCs can also potentially reach the remote atmosphere. Recent studies have shown that SOA may be more resistant to evaporation than previously thought, increasing the SOA lifetime and the distances it can be transported (Shrivastava et al., 2013, Vaden et al., 2011, Vaden et al., 2010, Zelenyuk et al., 2012,

Shrivastava et al., 2015). Furthermore, recent research has shown that while the lifetime of most biogenic SOA is shorter than their mechanical removal timescales, there exists a fraction of non-photolabile SOA that could potentially have long lifetimes and be effectively transported into the remote regions (Zawadowicz et al., 2020, O'Brien and Kroll, 2019). In summary, the MBL CCN budget is composed of primary ocean emissions (sea salt and primary organics), secondary aerosols derived from ocean biogenic VOCs, and aerosols transported remotely from the continents. The specific local variations of those sources,

their seasonality, and the exact mixing states of MBL aerosols are difficult to predict due to a lack of detailed chemical measurements of aerosol in remote marine regions, yet crucial for understanding cloud properties and radiative forcing in these areas.

This study focuses on chemical measurements of non-refractory aerosol and trace gas composition vertical profiles, from the Aerosol and Cloud Experiment in the Eastern North Atlantic (ACE-ENA), a U.S. Department of Energy (DOE) airborne

measurement campaign. The site of these measurements is the Azores archipelago in the Eastern North Atlantic (Figure 1), which is uniquely suited for characterization of both marine and long-range transported aerosol properties and various meteorological conditions favorable to both of these CCN composition regimes. Graciosa island in the Azores is also the location of a permanent DOE Atmospheric Radiation Measurement (ARM) user facility measurement site. Because the Azores straddle the boundary between subtropics and midlatitudes, they experience a wide range of meteorological conditions

throughout the year (Wood et al., 2015). There is a marked seasonality in the wind patterns near the Graciosa site: in the winter, there is a strong gradient of surface pressure between the Icelandic low and the Azores high pressure systems and the winds tend to have high average speeds and come from the southwest (Wood et al., 2015). In the summer, the Icelandic low disappears and the Azores high pressure system strengthens, which is associated with winds predominantly from the Northwest with lower average wind speeds (Wood et al., 2015). The high wind speeds are responsible for higher sea salt contributions to the ENA

CCN budget in the winter (Zheng et al., 2018). This pattern is also responsible for the seasonal peak in total cloud fraction,



which occurs the winter (Wood et al., 2015). This dynamic complexity makes it difficult to attribute sources of aerosols in the ENA region because the MBL air masses are continually diluted with free tropospheric air on a timescale of several days (Wood et al., 2015). As observed during the Clouds, Aerosol and Precipitation in the Marine Boundary Layer (CAP-MBL) campaign, marine air masses can have continental features due to the entrainment from the free troposphere (Wood et al.,

2015). Previous measurements on Pico island also indicate a seasonal summer peak in carbon monoxide due to long-range transport from North America (Val Martin et al., 2008). Additionally, single scattering albedo measurements during CAP-MBL indicate that the aerosols are more absorbing during springtime, also consistent with transport of North American biomass burning aerosols (Logan et al., 2014).

Other recent field measurements focusing on the aerosol chemistry of the ENA region include the NASA North Atlantic

Aerosols and Marine Ecosystems Study (NAAMES), which included both aircraft and shipborne observations focused on marine biological productivity in the region (Behrenfeld et al., 2019), and the African biomass burning-focused NASA Observations of Aerosols above Clouds and their Interactions (ORACLES), which included aircraft deployments of a suite of instruments, including the Aerosol Mass Spectrometer (AMS), in the Southeastern Atlantic. The AMS has a long field deployment history on a variety of platforms (Zhang et al., 2007), but particle composition measurements in remote marine

environments are of recent interest. Between 2011 and 2012, an AMS was deployed aboard the German research vessel Polarstern during four cruises in the Atlantic Ocean (Huang et al., 2017, Huang et al., 2018). The average aerosol composition was found to be ~50% sulfate and ~20% organic (Huang et al., 2018). The organic component was further analyzed with positive matrix factorization, and it was determined to be a mixture of primary and secondary (DMS- and amine-derived) marine aerosol and aerosol transported from North America (Huang et al., 2018). Between 2016 and 2017, AMS was also

deployed aboard the NASA DC-8 aircraft during the Atmospheric Tomography (ATom) missions in the remote atmosphere, including the North Atlantic region (Hodshire et al., 2019). In 2015, AMS was also deployed on aircraft as a part of the Network on Climate and Aerosols: Addressing Key Uncertainties in Remote Canadian Environments (NETCARE) project in the Canadian high Arctic (Abbatt et al., 2019, Willis et al., 2019). A unique feature of the ACE-ENA aircraft deployments is the seasonally-resolved measurements during summer and winter.

In this study, we focus on chemical characterization of aerosol and trace gases during ACE-ENA. We present average concentrations of sulfate, total organic, ammonium and nitrate aerosol components and mixing ratios of trace gases such as methanol, acetone, DMS, isoprene, toluene and benzene. In addition, concentrations of particle-phase methanesulfonic acid (MSA), an oxidation product of DMS, were derived using aircraft measurements and laboratory calibrations. We also discuss vertical profiles of these quantities in context of other G-1 measurements, meteorology and Hybrid Single-Particle Lagrangian

Integrated Trajectory model (HYSPLIT) (Stein et al., 2015) back-trajectories to elucidate the sources of ENA aerosols.





## 2 Experimental

The ACE-ENA campaign was conducted around the Azores archipelago in the Eastern North Atlantic from June 1, 2017 to February 28, 2018. During this period, the DOE G-1 research aircraft, based out of Lajes Field on Terceira Island (Figure 1), was deployed for two intensive operation periods (IOPs), consisting of flights around the Graciosa island ENA ARM research
site. The first aircraft deployment period occurred in the summer, from June 1, 2017 to July 31, 2017, and the second occurred in the winter, from January 1, 2018 to February 28, 2019. During the summertime deployment period, the aircraft completed 20 research flights, and during the winter, it completed 19 research flights (Supplementary Table S1). Supplementary Figure S1 shows flight tracks for all research flights. In general, the flight plans were focused on multiple L-shaped transits at different altitudes with the ENA site as the focal point, with occasional excursions to other cloud layers. Each research flight also
typically included at least one spiral profile through the atmosphere to characterize the boundary layer structure. The locations of spiral profiles are shown in Supplementary Figures S1C and S1D and included in Supplementary Table S1. This study uses the vertical profiles to explore the marine and continental influences on aerosol composition.

### 2.1 Aerosol Mass Spectrometry

An Aerodyne High-Resolution Time-of-Flight Aerosol Mass Spectrometer (DeCarlo et al., 2006) (abbreviated as AMS
thereafter) was deployed aboard the G-1 to measure non-refractory aerosol chemical composition. The AMS operated only in the standard "V" mass spectrometer mode with 13 s data averaging intervals and equal chopper open and closed periods of 3 s. The particle sizing mode was not used. Before and after the flights, air was diverted through a HEPA filter to remove the particulates, and these periods were used to account for gas-phase interferences with isobaric particulate signals. The AMS was regularly calibrated in the field using monodisperse ammonium nitrate particles quantified with a TSI condensation
particle counter (CPC), as described in the literature (Canagaratna et al., 2007, Jayne et al., 2000). A collection efficiency of unity is applied to data collected here, due to high acidity of the marine environment (Middlebrook et al., 2012). AMS data was processed using the ToF-AMS analysis toolkit Squirrel version 1.60N and ToF-AMS HR analysis toolkit Pika version 1.20N.

The AMS sampled from two G-1 inlets, an isokinetic aerosol inlet and a counter-flow virtual impactor (CVI) inlet. Sample
streams between two inlets were switched by the instrument operator aboard the aircraft based on cloud cover. This paper limits the discussion to data obtained during the isokinetic inlet sampling. It should be noted that this study is not sensitive to sea salt because AMS, which uses thermal desorption as a component of its ionization system, is not sensitive to refractory particle compositions.

Apart from regular calibrations with monodisperse ammonium nitrate, AMS was also calibrated for MSA in the laboratory
after the campaign. The details of this calibration are discussed in the Supplementary Information.



## 2.2 Trace gas mass spectrometry

An Ionicon quadrupole high-sensitivity Proton-Transfer-Reaction Mass Spectrometer (abbreviated as PTR-MS thereafter) was used to measure selected gas-phase VOC concentrations. The PTR-MS was run in ion monitoring mode in which signals of a limited number of pre-selected m/z values are sequentially measured, with one measurement cycle taking 3.5 s. Drift tube

temperature, pressure, and voltage were held at 60ºC, 2.22 hPa, and 600 V, respectively. The PTR-MS sampled air through a dedicated inlet that consisted of approximately 6" of 1/4" OD stainless steel, followed by approximately 46" of 1/4" OD Teflon tubing, including a Teflon filter, and 36" of 1/16" OD PEEK tubing. To assess the PTR-MS background, air was periodically diverted through a stainless-steel tube filled with a Shimadzu platinum catalyst heated to 600ºC, which removed VOCs from the airstream without perturbing the water vapor content. The PTR-MS was calibrated by introducing known concentrations

of calibration gases into the instrument with variable dilution by VOC-free air. Two different calibration tanks were used; one in the field and one in the lab before and after the IOPs. Much of the summer IOP PTR-MS data was affected by a loss of sensitivity resulting from a faulty electrical connection in the quadruple. Additionally, DMS backgrounds throughout the summer campaign were found to be elevated, even with the flow diverted through the catalyst. This could be due to an isobaric interference or incomplete removal of DMS by the catalyst. Absolute concentrations of DMS, especially during the summer

intensive period, are biased high due to these factors and should be regarded as largely qualitative.

## 2.3 Supporting measurements

Ozone was measured with a Thermo Scientific Model 49i ozone analyzer based on measurement of UV absorption at 254 nm. The instrument was regularly calibrated in flight by displacement of known quantities of ozone and zeroed in flight using ozone-scrubbed ambient air. CO was measured using Los Gatos Research CO-$N_2$O-$H_2$O analyzer based on cavity-enhanced

near-IR absorption and was also calibrated regularly in flight. Refractory black carbon concentration was measured using DMT Single-particle Soot Photometer (SP2). The SP2 was calibrated at the beginning, during and at the end of the deployment using fullerene soot in order to more closely mimic the morphology of ambient black carbon. CCN concentrations were measured with the Droplet Measurement Technologies CCN-200 Cloud Condensation Nuclei Counter. A CCN counter draws ambient aerosol through a column with supersaturated water vapor, where they can activate into cloud droplets. The instrument then

sizes and counts activated ambient aerosol as a function of supersaturation (Roberts and Nenes, 2005). A two column CCN counter was used on the G-1, with one column at 0.1% supersaturation and the other at 0.3% supersaturation. The instrument was calibrated with size-selected ammonium sulfate. G-1 position and altitude and wind direction and velocity were measured using Aventech AIMMS-20 probe.



## 3 Results

### 3.1 Background aerosol and trace gas composition in the Eastern North Atlantic

#### 3.1.1 Aerosol chemistry

Figure 2 and Table 1 summarize AMS measurements obtained during the 39 ACE-ENA research flights. In the summer MBL, mean loadings for organic, sulfate, ammonium and nitrate were 0.18, 0.55, 0.05 and 0.01 $\mu g\ m^{-3}$, respectively (Table 1). These are low concentrations, illustrating clean conditions in the remote ENA region. The low abundance of nitrate is expected given minor influence of anthropogenic pollution over the remote North Atlantic. Low ammonium is also expected, given the distance from ammonia emissions. In the summer free troposphere, mean loadings of the same species were 0.12, 0.18, 0.03 and 0.01 $\mu g\ m^{-3}$, respectively (Table 1). In the MBL, mean winter loadings for organic, sulfate, ammonium and nitrate were 0.04, 0.11, 0.01 and 0.003 $\mu g\ m^{-3}$, respectively (Table 1). In the free troposphere, mean winter loadings for the same species were 0.03, 0.06, 0.01 and 0.002 $\mu g\ m^{-3}$, respectively (Table 1). Apart from low average abundance, these measurements also show strong seasonality with significantly lower concentrations of all aerosol species observed in the winter campaign. For both organic and sulfate, abundances during the summer were four times larger than in the winter. For comparison, mean organic, sulfate, ammonium and nitrate concentrations measured in GoAmazon 2014/15 in the central amazon basin, a remote continental location, during the wet season were 0.91, 0.16, 0.05, and 0.02 $\mu g\ m^{-3}$ respectively (Shilling et al., 2018). AMS measurements of organic, sulfate, ammonium and nitrate from the NEAQS 2002 campaign measured from a ship in the vicinity of the NE USA, a likely source region of transported aerosol, were 5.0, 2.1, 0.65 and 0.30 $\mu g\ m^{-3}$ respectively (Zhang et al., 2007, de Gouw et al., 2005). AMS measurements of the same species aboard research ship Polarstern during the North Atlantic transect along the coast of Europe and Africa were 0.53, 1.38, 0.29 and 0.09 $\mu g\ m^{-3}$, respectively, in the spring and 0.47, 0.76, 0.20 and 0.07 $\mu g\ m^{-3}$, respectively, in the winter (Huang et al., 2018). Thus, the AMS measurements during ACE-ENA represent some of the cleanest conditions measured with AMS.

Figure 3 shows a flight-by-flight summary of AMS-derived MSA measurements obtained using a laboratory calibration, as outlined in the Experimental section. MSA measurements shown in Figure 3 are for altitudes below 1000 m, which are expected to show the strongest marine influence. Campaign averages for MSA are also summarized in Table 1. Average MSA concentrations for altitudes < 1000 m during the summer period were 0.021 $\mu g\ m^{-3}$, compared to 0.002 $\mu g\ m^{-3}$ during the winter. The range of concentrations measured during ACE-ENA is comparable to other MSA measurements over the Eastern North Atlantic, for example, Huang et al. (2017) reports 0.04 $\mu g\ m^{-3}$ in the spring and 0.01 $\mu g\ m^{-3}$ in the fall, measured with an AMS aboard a research ship in the North Atlantic. MSA concentrations strongly vary by season as well; MSA is about 10 times less abundant during the winter compared to the summer. In Figure 3C and 3D, total particulate MSA loadings are expressed as fractions of AMS sulfate signal as described in the Supplementary Information. Even during the summer, MSA accounts for a small (<10%) fraction of the total particulate sulfate. The small fraction of particulate MSA does not, however, indicate that DMS is not a significant source of sulfate mass in the region, as DMS oxidation also produces inorganic sulfate.



Despite strong seasonality in abundance, relative contributions of non-refractory aerosol compositions are similar between summer and winter. Figure 4 shows relative contributions of sulfate, organic, ammonium and nitrate to the total non-refractory aerosol budget measured by the AMS during both measurement periods. In both IOPs, the fractional contribution of organics, sulfate, ammonium and nitrate were 23, 69, 6 and 1%, respectively. Sulfate is the dominant contribution to the non-refractory

aerosol mass in the ACE-ENA region. This in in contrast to the dominance of organics at most continental sites in the northern hemisphere mid-latitudes (Zhang et al., 2007). We observe a trend of decreasing contribution of sulfate with increasing altitude. Sulfate contributes 51-58% to the total non-refractory aerosol at altitudes between 1000 m and 3000 m and 68-70% at altitudes below 1000 m. Figure 4 also shows an estimate of average relative MSA contributions to sulfate and organic components, which do not exceed 1% and 2%, respectively. While MSA is a useful tracer for marine biogenic influence, it does not account

for majority of the particulate sulfate mass in the boundary layer.

Unlike sulfate, there is no strong trend of decreasing ammonium with altitude and as a result, boundary layer sulfate aerosol is more acidic than sulfate in the free troposphere. AMS measurements can be used to calculate the degree of aerosol neutralization as:

$$\text{Degree of neutralization} = \frac{[NH_4]/18}{\frac{2[SO_4]}{96} + \frac{[NO_3]}{62} + \frac{[Cl]}{35.5}}$$

Neutralization below 1 indicates acidic aerosol. As shown in Figure 9, it is especially clear during the summer that the surface-derived marine aerosol tends to be strongly acidic, but the higher altitude aerosol tends to be more neutralized, further supporting two distinct sources of sulfate at different altitudes. Note that this analysis is based solely on quantities measured

by the AMS and therefore neglects certain common marine cations such as Na$^+$ or K$^+$. Chloride salts of Na or K will also be undetected by the AMS. Due to the acidity of marine boundary layer sulfate, the bulk biogenic sulfate is likely sulfuric acid, which is produced from oxidation of SO$_2$, another oxidation product of DMS (Hoffmann et al., 2016). Use of thermodenuder in front of an aerosol sizing instrument aboard the G-1 aircraft during the ACE-ENA campaign was used to infer that the AMS observations are of non-sea salt submicron marine aerosol and any coarse sea salt is externally mixed and undetected by the

AMS.

Relative contributions of different functional groups differentiated by AMS to the total bulk organic loading are also indicated in Figure 4. In all cases, CxHyOz components contribute over 50% of total organic aerosol, as would be expected for photochemically aged aerosol transported into the remote region. Figure 5 shows a further overview of organic aerosol chemistry during the summer, when organics were most abundant. O:C and H:C ratios were computed from AMS

measurements using the "Improved Ambient" method of Canagaratna et al. (2015) and averaged using a 10-point moving average. The O:C ratio of 1.1 for MBL and 1.0 for free troposphere suggests that the organic aerosol is highly processed by the time is sampled at the ACE-ENA region. For comparison, the mean O:C and H:C ratios during GoAmazon 2014/15 were 0.6 and 1.65 during the wet season (Shilling et al., 2018). No strong trends in elemental ratios are observed for the two altitude





bins considered in Figure 5A and 5B, < 1000 m and 1000 m - 3000 m. This suggests a similar degree of oxidation of the
organic aerosol throughout the marine boundary layer, further pointing to a lack of strong local sources of organic aerosol.

To further test this, two representative research flights, RF #9 and #19 were selected to represent conditions with lower (RF
#9) and higher (RF #19) influence from long-range transport. The vertical profiles of total AMS organic loading from those
two research flights are shown in Figure 6. During RF #9, MSA accounts for 3% of the total particulate organic signal, but
during RF #19, its contribution is less than < 1% (Figure 6C). However, the distributions of O:C and H:C ratios on those two
days are very similar (Figures 6D and 6E). This suggests two likely possibilities for the source of organic aerosol in the ENA
region: (1) the majority of the boundary layer particulate organic is oxidized continental emissions that were transported and
are now mixed into the boundary layer, (2) fresh ocean primary organic aerosol that is highly oxidized and indistinguishable
from long-range transported organic aerosol by O:C and H:C ratios. During the International Chemistry Experiment in the
Arctic Lower Troposphere (ICEALOT) O:C of submicron particles was quantified using both FTIR and AMS aboard a
research ship (Frossard et al., 2011, Russell et al., 2010). The average O:C during the cruise was 0.94, and 1.07 for the North
Atlantic leg (Russell et al., 2010). This high O:C ratio was linked to an abundance of highly oxidized hydroxyl functional
groups similar to those found in biogenic carbohydrates found in sea water (Russell et al., 2010). Bubble bursting on the ocean
surface was identified as a likely source of such primary aerosols (Russell et al., 2010).

Figure 6 also shows aerosol $pH_F$ (i.e., pH based on only the free-H+ molality, (Pye et al., 2020)) calculated using the MOSAIC
aerosol model (Zaveri et al., 2008) and the web-based E-AIM model IV (Friese and Ebel, 2010, Wexler and Clegg, 2002) for
the two limiting cases of local vs. remote emissions, respectively. This provides a more accurate constraint on the aerosol pH
than the neutralization obtained from AMS, but it agrees with the earlier conclusion that the submicron aerosol in the MBL at
ENA is very acidic. In the case of clean MBL, during RF #9, pH is between 0 and -2, depending on the presence of liquid
water in the particles. Even during RF #19, which contained a clear remote transport plume, pH is 0 everywhere outside of the
plume. This raises the possibility of a third source of organic aerosol at ENA, fresh SOA produced through acid-catalyzed
reactive uptake of oxidized isoprene and monoterpene emitted from the ocean.

### 3.1.2 Trace gas chemistry

Figure 7 and Table 2 summarize the PTR-MS measurements for trace gases during ACE-ENA. The most abundant trace gas
is methanol, with average concentrations around 2 ppb at all altitudes during the summer (Table 2). In the winter, the average
methanol concentration was 0.6 ppb, and the average DMS concentration was 0.1 ppb (Table 2). Isoprene, benzene and toluene
concentrations are very low throughout the campaign, with average concentrations below 0.5 ppb for isoprene and below 0.1
ppb for benzene and toluene, which is near the detection limit. The measurements of these anthropogenic and biogenic VOCs
again indicate that emissions of SOA forming precursors in the region are low. The three major trace gas components (DMS,
methanol and acetone) all show seasonality, but the reduction in concentration from summer to winter is 3-4 times for methanol
and acetone and 20-fold for DMS. Note that there is a potential interference for summer DMS, which produced higher absolute
concentrations than expected, though we expect trends in relative concentration to be accurate.





## 3.2 Vertical profiles of aerosol and trace gases

Vertical profiles for each of the research flights are summarized in Supplementary Table S1. Most flights included at least two vertical profiles, and if there were multiple vertical profiles per flight, they were averaged together. Vertical profile locations

are shown in Supplementary Figure S1; the majority were located close to the coast of Graciosa Island. Figure 8 shows vertical profiles of aerosol chemistry measured with the AMS. Examination of vertical profiles can reveal the boundary layer structure and aerosol sources. Because all the vertical profiles were carried out over the ocean, the surface sources are marine in origin. The vertical profiles of organics and nitrate are relatively constant on most flights, indicating that these species are well-mixed through the troposphere. Organics display a slight increase in concentration at high altitudes in the winter, though this layer is

relatively weak. Ammonium aerosol components both have weak marine sources and a stronger influences at > 1000 m in the summer. Similar to organics, ammonium shows weak high altitude layers above 2000 m in the winter, likely representing transport from North America. Sulfate aerosol has two distinct sources, a marine source seen below 1000 m and a high altitude source above 1000 m in both seasons. Figure 9 shows additional vertical profiles of quantities derived using AMS aerosol chemistry measurements. MSA shows a strong surface source during the summer measurement period, but falls below the

detection limit during the winter measurement period. Figure 10 shows vertical profiles of refractory black carbon (rBC) measured by the SP-2. Consistently with clean, remote conditions, concentrations of rBC are low in both measurement periods, but during the summer several profiles show very clear layers at altitudes higher than 1000 m. Profiles of rBC are consistent with occasional transport of polluted air from the continent at high altitude to the region.

Figure 11 shows vertical profiles of four of the trace gases measured with the PTR-MS. Two of these gases, DMS and isoprene

are expected to have local sources: marine biogenic emissions in case of DMS and both marine and island biogenic emissions for isoprene. The source for DMS emissions is decidedly marine, but the isoprene profiles don't show a strong surface source, suggesting a lack of local isoprene emissions. Phytoplankton have been shown to emit isoprene in the laboratory (Colomb et al., 2008, Meskhidze et al., 2015, Shaw et al., 2010, Exton et al., 2013), and trace isoprene and its oxidation products have been detected in ocean research cruises (Hu et al., 2013, Kim et al., 2017, Mungall et al., 2017, Hackenberg et al., 2017).

Generally, the phytoplankton species that emit isoprene are different from those that are primary DMS producers, and their respective geographical distributions are anti-correlated (Dani and Loreto, 2017). Methanol and acetone are both expected to be tracers for continental transport, and they tend to show layers at altitudes higher than 1000 m during both measurement periods. Figure 12 shows vertical profiles of two additional atmospheric gases whose concentrations were also measured aboard the G-1, carbon monoxide and ozone. Both of those gases are tracers for long-range transport of anthropogenic pollution

and/or biomass burning. They show similar high altitude layers to methanol and acetone, but CO in particular is less susceptible to wet removal by clouds and precipitation because of its low water solubility. Summertime marine boundary layer concentrations for carbon monoxide and ozone are 60 - 100 ppb and 10 - 40 ppb, respectively. During the winter, however, their concentrations are consistently higher, 80 - 140 ppb and 20 - 50 ppb, respectively. This can indicate both continental influence and more efficient removal of CO during the summer due to photochemistry. Recently, concentrations of hydroxyl



radical have been quantified during the summer and winter in the remote troposphere, indicating a column average for the
Northern Hemisphere of $4.4 \times 10^6$ molecules $cm^{-3}$ in the summer and $0.9 \times 10^6$ molecules $cm^{-3}$ in the winter, which could
explain the seasonal differences in photochemical removal (Travis et al., 2020). In the Amazon, the CO background was found
to be about 80 ppb absent any urban influence (Shilling et al., 2018). Similar background CO concentration was also found in
the remote Southeast Atlantic (Zuidema et al., 2018).

Figure 13 shows vertical profiles of CCN concentrations measured with the CCN counter aboard the G-1. During both seasons,
but especially in the summer, CCN profiles corelate with high altitude layers of organic, ammonium and rBC, which are
markers for long-range transport of aerosols, as described above. This trend is apparent for both 0.1% and 0.3%
supersaturations. Campaign-wide correlations between aerosol chemical composition and CCN concentrations are also shown
in Supplementary Figures S3 and S4. At both supersaturations, CCN concentration correlate best with ammonium at high

altitudes (>1000 m), $R^2 = 0.8$ for 0.1% supersaturation and $R^2 = 0.7$ for 0.3% supersaturation, which tends to be transported
from the continents.

## 4 Discussion

Examination of average concentrations of trace gases and aerosols, their vertical profiles and external factors such as
meteorology, back-trajectories and biological productivity reveals an interplay of local and transported emissions giving rise

to CCN in the ENA region. Two tracers of biological ocean productivity were measured during ACE-ENA: DMS and
particulate MSA. Both of these tracers are prominent in the marine boundary layer, but generally only in the summer. The
synchronization in concentrations of DMS and MSA is expected, as MSA is produced through atmospheric oxidation of DMS.
In the marine boundary layer, DMS is a by-product of metabolism of some phytoplankton species, such as coccolithophores
and dinoflagellates (Keller et al., 1989). Because its production is biogenic in nature, it undergoes seasonal cycles, which tend

to be strongly latitude-dependent (Galí and Simó, 2015, Polimene et al., 2011). At polar and sub-polar latitudes, which are
light-limited in the winter, concentrations of ocean DMS peak synchronously with phytoplankton biomass, but at subtropical
latitudes, peak DMS emissions lag behind peak biomass by as much as a few months (Galí and Simó, 2015, Polimene et al.,
2011, Lana et al., 2011). The cause of this lag is the way seasonality in solar irradiation drives diversity of phytoplankton taxa,
with the DMS-producing species preferring more irradiated conditions (Galí and Simó, 2015, Polimene et al., 2011).

Supplementary Figure S5 shows distributions of ocean chlorophyll-a measured by MODIS Aqua satellite (NASA Goddard
Space Flight Center, 2018) during the two intensive periods during ACE-ENA. While the summer coincides with greater
biological productivity at the sub-polar latitudes North of the Azores, the chlorophyll-a concentrations at the ENA site are
higher in the winter (0.27 mg $m^{-3}$) than in the summer (0.12 mg $m^{-3}$), which is an opposite trend to the aircraft DMS and MSA
measurements. This could be due to the lag in seasonal production of DMS in the Azores, which straddle midlatitude and

subtropical regimes. The anti-correlation between phytoplankton biomass and ocean DMS concentrations has been shown to
persist in the -20º - 40º latitude band in both hemispheres, which includes the Azores (Lana et al., 2011). The seasonal surface



DMS concentration has been shown to peak in the summer in the Eastern North Atlantic using a global model (Kloster et al., 2006). Previous measurements of surface seawater DMS concentrations compiled in the Global Surface Seawater DMS Database (saga.pmel.noaa.gov/dms/) (Kettle et al., 1999) also indicate higher concentration around the Azores in the summer

(mean 2 ± 2 nM/L) than in the winter (0.5 ± 0.4 nM/L) (Supplementary Figure S6). An alternative explanation is transport of DMS-rich air masses North of the site, but the HYSPLIT (Stein et al., 2015) back-trajectories shown in Supplementary Figure S7 do not suggest strong transport from the North at low altitudes where DMS signal is most prominent.

Local production of MSA and sulfate from oceanic emissions is a major source of aerosol mass in the summer. Sulfate is highly hygroscopic and therefore this aerosol will be efficient as CCN, which explains higher on average CCN concentrations

close to the ocean surface in the summer (Figure 13). There is evidence from analysis of ENA site data that accumulation mode particles entrained from the free troposphere, as well as growth of Aitken-mode particles into accumulation mode in the MBL are the largest contributors to the CCN budget in the ENA region (Zheng et al., 2018). In particular, the latter source can represent as much as 60% of the CCN budget in the summer (Zheng et al., 2018). Aiken-mode particles, in this case, are entrained from the free troposphere and they continue to grow by condensation of DMS oxidation products, such as MSA and

sulfuric acid into CCN-relevant sulfate particles (Zheng et al., 2018). Annual mean contributions of sea salt to accumulation and Aiken modes have been found to be 21% and less than 10%, respectively (Zheng et al., 2018), which suggests that AMS is sensitive to the majority of the ENA CCN budget composition.

Vertical profiles presented in Figures 8, 9, and 10 show strong stratification in aerosol chemistry. Below ~1000 m, the particles are MSA-influenced and strongly sulfate-dominated and acidic; at 1000 m - 3000 m, the particles contain no MSA, are more

neutralized (though still acidic) and can contain organic, ammonium and black carbon. HYSPLIT back-trajectories in Supplementary Figure S7 corroborate the origin of low-altitude air masses as primarily marine. The high altitude features are also consistent with long-rage transport of North American continental emissions. Research flight #19 on July 19, 2017 shows the strongest continental transport influence at altitudes between 1000 m and 2500 m. The peak in aerosol abundance, organic and black carbon concentrations and higher pH (Figure 6) is accompanied by a peak in CO, methane and acetone concentrations

(Figures 11 and 12). Those characteristics suggest a biomass burning plume, but the lack of AMS m/z 60 levoglucosan marker points to highly processed biomass burning aerosol or anthropogenic emissions. In order to constrain the source of the emissions, HYSPLIT back-trajectory analysis of this flight was carried out as shown in Figure 14. An ensemble of 14-day back-trajectories was generated from the site of the first vertical profile (Supplementary Table S1) at 2000 m altitude. The back-trajectories reach > 500 m altitude over the East of the United States on July 12 - 15. The Fire Inventory from NCAR

(FINNv1) (Wiedinmyer et al., 2011) was then used to provide locations of major fires on these four days, as shown in Supplementary Figure S8. Two fire clusters on those days occur in Southwestern Canada and Southeastern United States. The Southeastern U.S. location in South Carolina, where FINN predicts the highest biomass burning emissions was used to start 10-day forward-trajectories in HYSPLIT in Figure 14. Those reach the Azores at the correct time and altitude, which suggest that Southeastern U.S. could be the location for biomass burning emissions seen in the ENA region on July 19, 2017. A similar

back-trajectory analysis was carried out for the Southwestern Canadian fire location (Supplementary Figure S9), and those



trajectories do not appear to reach the correct location. However, uncertainties in trajectory analysis and factors such as rapid mixing of free tropospheric and MBL air (Wood et al., 2015) do not allow ruling it out as a source completely. Anthropogenic emissions such as traffic also cannot be ruled out completely as a source of July 19, other polluted layers at ENA, or the highly oxidized background organic aerosol. Such polluted layers were observed relatively frequently during the campaign and they

were often accompanied by large increases in CCN concentrations (Figure 13). This shows that periodic long-range transport of pollution into the ENA region can significantly perturb the local CCN budget.

Long-range transported aerosols (organic, ammonium, rBC) and reactive gases (methanol, acetone) at ENA show strong seasonality with high concentrations in the summer and low in the winter. This can be partially attributed to the seasonality of North American wildfires, which occur in the summer and fall. However, Figure 12 suggests an opposite trend for CO, a non-

reactive, water-insoluble gas, which on average shows higher concentrations in the winter. Analysis of three years of data at the ENA ARM site suggests a peak of CO and ozone concentrations occurring in the spring (March) and a minimum during the summer (July) (Zheng et al., 2018). This trend coincides with the CO and ozone seasonality seen during ACE-ENA. One explanation is that CO undergoes more photolytic destruction in the summer due to higher OH concentrations. Another is that continental transport into the Azores might actually be more prominent during the winter, but the particles and reactive gases

are may be locally scavenged by clouds. Supplementary Figure S5 shows MODIS Aqua observations of cloud fraction over the North Atlantic during the two intensive observation periods during the ACE-ENA campaign. The mean cloud fraction over the Azores is 0.56 and 0.77 during summer and fall, respectively. Additionally, Supplementary Figure S10 shows that on average, the winds were stronger in the winter, and the HYSPLIT back-trajectories in Supplementary Figure S7 show more long-range transport during the winter at all altitudes. The seasonal trend in CO can also be attributed to local emissions to

some extent, as higher emissions in the winter coincide with residential heating and power use. EDGAR-HTAP V2 (https://edgar.jrc.ec.europa.eu/htap_v2/) gridded emissions inventory (Janssens-Maenhout et al., 2015) was used to estimate local CO emissions in the Azores (Supplementary Figure S11), showing increased emissions in the power and residential sectors. However, local Azores anthropogenic emissions are still rather low, and the trends in local emissions and transported pollution are likely superimposed.

**5 Conclusion**

During the ACE-ENA campaign, 39 research flights in the vicinity of the Graciosa Island in the Azores were carried out during two seasons, summer, 2017 and winter, 2018. Aerosol and trace gas chemistry were characterized using the AMS and PTR-MS, as well as a suite of other sensors. AMS measurements reveal clean conditions with < 1 $\mu$g m$^{-3}$ non-refractory aerosol abundance and strong seasonality, where the summer aerosol concentrations were ~4 times higher than in the winter. Trace

reactive gas concentrations were often < 1 ppb throughout the campaign, with similar seasonality. Particles sampled in the MBL showed contributions from MSA, which is a tracer for ocean biogenic activity. The summertime clean MBL at ENA is characterized by secondary sulfuric acid aerosol produced through DMS oxidation and some highly oxidized organic aerosol,



which could be entrained from the free troposphere or ejected from the ocean. Secondary sulfate aerosol is also hygroscopic, which leads to high baseline concentration of CCN above the ocean (100 cm$^{-3}$ for 0.1% supersaturation and 200 cm$^{-3}$ for 0.3%

supersaturation). In the winter, less DMS is emitted due to seasonal shifts in ocean productivity, and consequently aerosol and CCN concentrations in the boundary layer also decrease (~50 cm$^{-3}$ for both supersaturations sampled).

Examination of vertical profiles and HYSPLIT back-trajectories revealed numerous remote transport events that influenced the aerosol sampled in the free troposphere, above 1000 m. Both anthropogenic pollution and North American wildfires are a likely sources, with especially high emissions observed on July 19, 2017. Such long-range transport events were accompanied

by large plumes of CCN in the free troposphere (Figure 13). The July 19, 2017 event was in particular traced to possible fires in the North American SEUS region and it was associated with a large plume of CCN (200-300 cm$^{-3}$ for 0.1% supersaturation and 300-400 cm$^{-3}$ for 0.3% supersaturation). This work shows that the CCN budget in the MBL in the Eastern North Atlantic region is a result of complex interactions between long-rage transport of North American wildfire and anthropogenic emissions and the local marine biogenic sources.


**Acknowledgements**

The authors thank the G-1 flight and ground crews for supporting the ACE-ENA campaign. This research was supported by the Office of Science of the U.S. Department of Energy (DOE) as part of the Atmospheric System Research (ASR) Program

via Grant KP1701000/57131.

**Author contributions**

MAZ, KS, JL, MP and JES collected the AMS and PTR-MS data aboard the G-1, FM, AS, SS, YW and JW collected and

analyzed other airborne data used in this paper, MAZ and JES analyzed the AMS and PTR-MS data, RAZ provided the MOSAIC model results, MAZ wrote the paper with contributions and advice from all co-authors.

**Data availability**

All ARM datasets used in this paper are publicly available on the ARM website (arm.gov/data).

**Competing interests**

The authors declare that they have no competing interests.



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



**Table 1:** Summary of AMS measurements aboard the G-1 during ACE-ENA.

| | IOP 1, summer | | IOP 2, winter | |
|---|---|---|---|---|
| | *mean* | *median* | *mean* | *median* |
| **Altitudes < 1000 m** | | | | |
| | *mean* | *median* | *mean* | *median* |
| **Organic** | 0.175 | 0.142 | 0.040 | 0.039 |
| **Sulfate** | 0.548 | 0.529 | 0.111 | 0.090 |
| **Ammonium** | 0.051 | 0.050 | 0.009 | 0.007 |
| **Nitrate** | 0.005 | 0.004 | 0.003 | 0.003 |
| **MSA** | 0.021 | 0.019 | 0.002 | 0.001 |
| **Altitudes 1000 m - 3000 m** | | | | |
| | *mean* | *median* | *mean* | *median* |
| **Organic** | 0.115 | 0.088 | 0.032 | 0.029 |
| **Sulfate** | 0.182 | 0.129 | 0.056 | 0.042 |
| **Ammonium** | 0.029 | 0.019 | 0.005 | 0.003 |
| **Nitrate** | 0.005 | 0.003 | 0.002 | 0.002 |
| **MSA** | 0.003 | 0.002 | 0.001 | 0 |
| **All altitudes** | | | | |
| | *mean* | *median* | *mean* | *median* |
| **Organic** | 0.146 | 0.116 | 0.036 | 0.034 |
| **Sulfate** | 0.375 | 0.328 | 0.078 | 0.056 |
| **Ammonium** | 0.040 | 0.030 | 0.007 | 0.004 |
| **Nitrate** | 0.005 | 0.004 | 0.003 | 0.002 |
| **MSA** | 0.012 | 0.008 | 0.001 | 0.001 |



**Table 2:** Summary of PTR-MS measurements aboard the G-1 during ACE-ENA.

| | IOP 1, summer | | IOP 2, winter | |
|---|---|---|---|---|
| **Altitudes < 1000 m** | | | | |
| | *mean* | *median* | *mean* | *median* |
| **Methanol** | 1.65 | 1.45 | 0.38 | 0.38 |
| **Acetone** | 1.14 | 0.95 | 0.25 | 0.23 |
| **DMS** | 3.31 | 2.81 | 0.20 | 0.18 |
| **Isoprene** | 0.21 | 0.03 | 0.11 | 0.08 |
| **Benzene** | 0.05 | 0.03 | 0.06 | 0.05 |
| **Toluene** | 0.13 | 0.07 | 0.03 | 0.03 |
| **Altitudes 1000 m - 3000 m** | | | | |
| | *mean* | *median* | *mean* | *median* |
| **Methanol** | 2.51 | 2.28 | 0.65 | 0.63 |
| **Acetone** | 1.67 | 1.34 | 0.36 | 0.32 |
| **DMS** | 2.06 | 1.74 | 0.07 | 0.05 |
| **Isoprene** | 0.31 | 0.05 | 0.13 | 0.10 |
| **Benzene** | 0.05 | 0.03 | 0.05 | 0.05 |
| **Toluene** | 0.11 | 0.07 | 0.03 | 0.03 |
| **All altitudes** | | | | |
| | *mean* | *median* | *mean* | *median* |
| **Methanol** | 2.06 | 1.85 | 0.57 | 0.54 |
| **Acetone** | 1.39 | 1.14 | 0.32 | 0.29 |
| **DMS** | 2.67 | 2.34 | 0.12 | 0.09 |
| **Isoprene** | 0.26 | 0.04 | 0.13 | 0.09 |
| **Benzene** | 0.05 | 0.03 | 0.05 | 0.05 |
| **Toluene** | 0.12 | 0.07 | 0.03 | 0.03 |





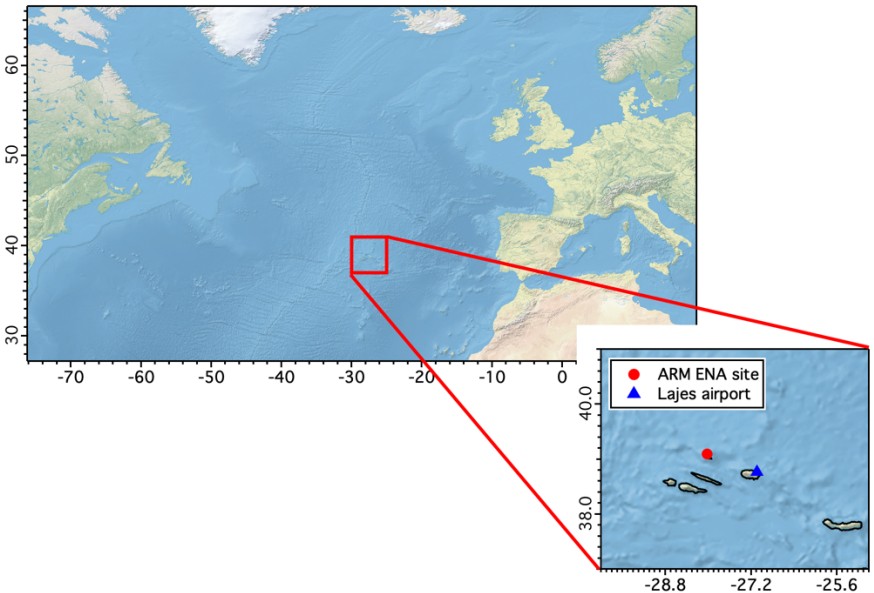

**Figure 1.** Geographical location of the Azores archipelago, the site of the ACE-ENA campaign. The map was created using
public domain map data on Natural Earth (naturalearthdata.com) and the GSHHG Database (ngdc.noaa.gov/mgg/shorelines/).



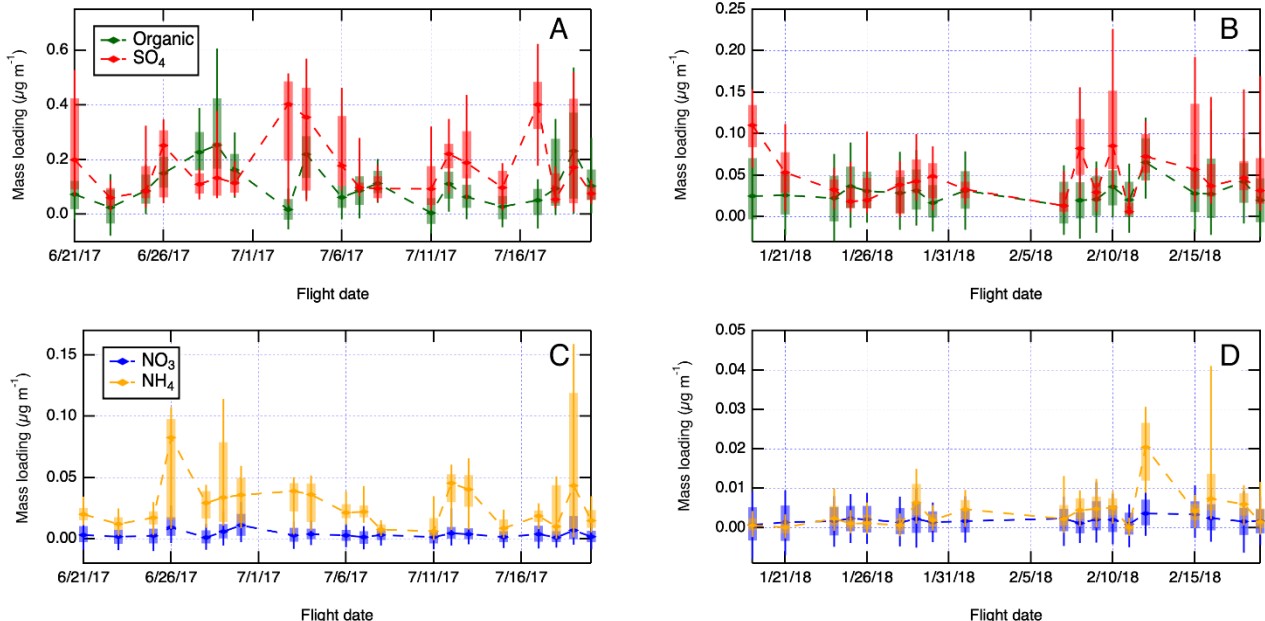

**Figure 2.** Summary of aerosol chemistry measurements acquired with the HR-ToF-AMS aboard the G-1 aircraft during ACE-ENA. Boxes are centered on daily medians, box boundaries extend between 25th and 75th percentile, and whiskers extend between 10th and 90th percentile. (A) and (C) IOP 1 measurements. (B) and (D) IOP 2 measurements. (A), (B), (C) and (D) are for altitude window 1000 m - 3000 m.






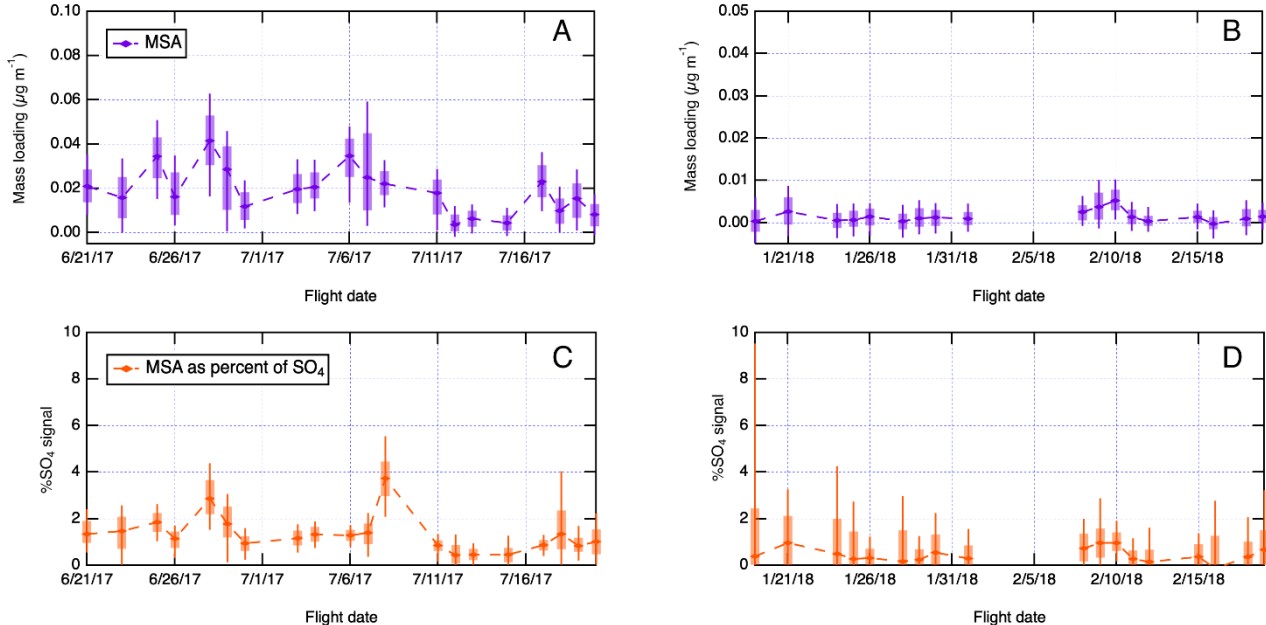

**Figure 3.** Summary of methanesulfonic acid (MSA) measurements derived from the AMS mass spectra collected during ACE-
ENA and laboratory calibrations as described in the Experimental section, expressed as total particle-phase MSA (A and B)
and percentage of total AMS sulfate signal (C and D). Boxes are centered on daily medians, box boundaries extend between
$25^{th}$ and $75^{th}$ percentile, and whiskers extend between $10^{th}$ and $90^{th}$ percentile. (A) and (C) IOP 1 measurements. (B) and (D)
IOP 2 measurements. (A), (B), (C) and (D) are for altitude window < 1000 m.




**Figure 4.** Relative contributions of four non-refractory components of aerosol chemistry: organic, sulfate, ammonium and nitrate to boundary layer AMS observations shown in Figure 2 and Table 2 at two different altitude ranges. The hatched portions of organic and sulfate observations represent contributions of MSA. Also shown are relative contributions of different functional groups to the total organic budget. (A) 1000 m - 3000 m, IOP 1. (B) 1000 m - 3000 m, IOP 2. (C) < 1000 m, IOP 1. (D) < 1000 m, IOP 2.





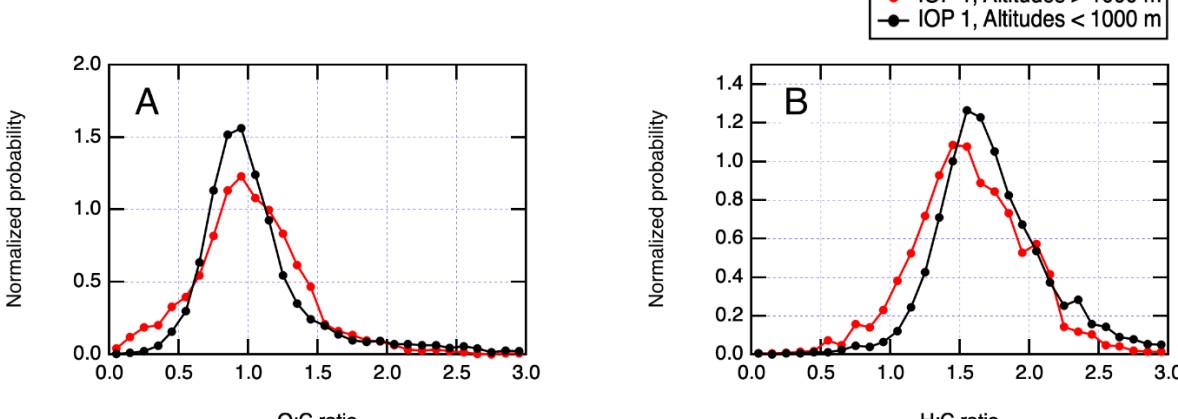

**Figure 5.** Overview of organic chemistry for IOP 1 using AMS measurement periods where [Org] > 0.15 µg/m³. All elemental ratios were calculated using the Canagratna et al. (2015) method. (A) AMS O:C ratios for boundary layer and free troposphere altitudes. (B) AMS H:C ratios for boundary layer and free troposphere altitudes.





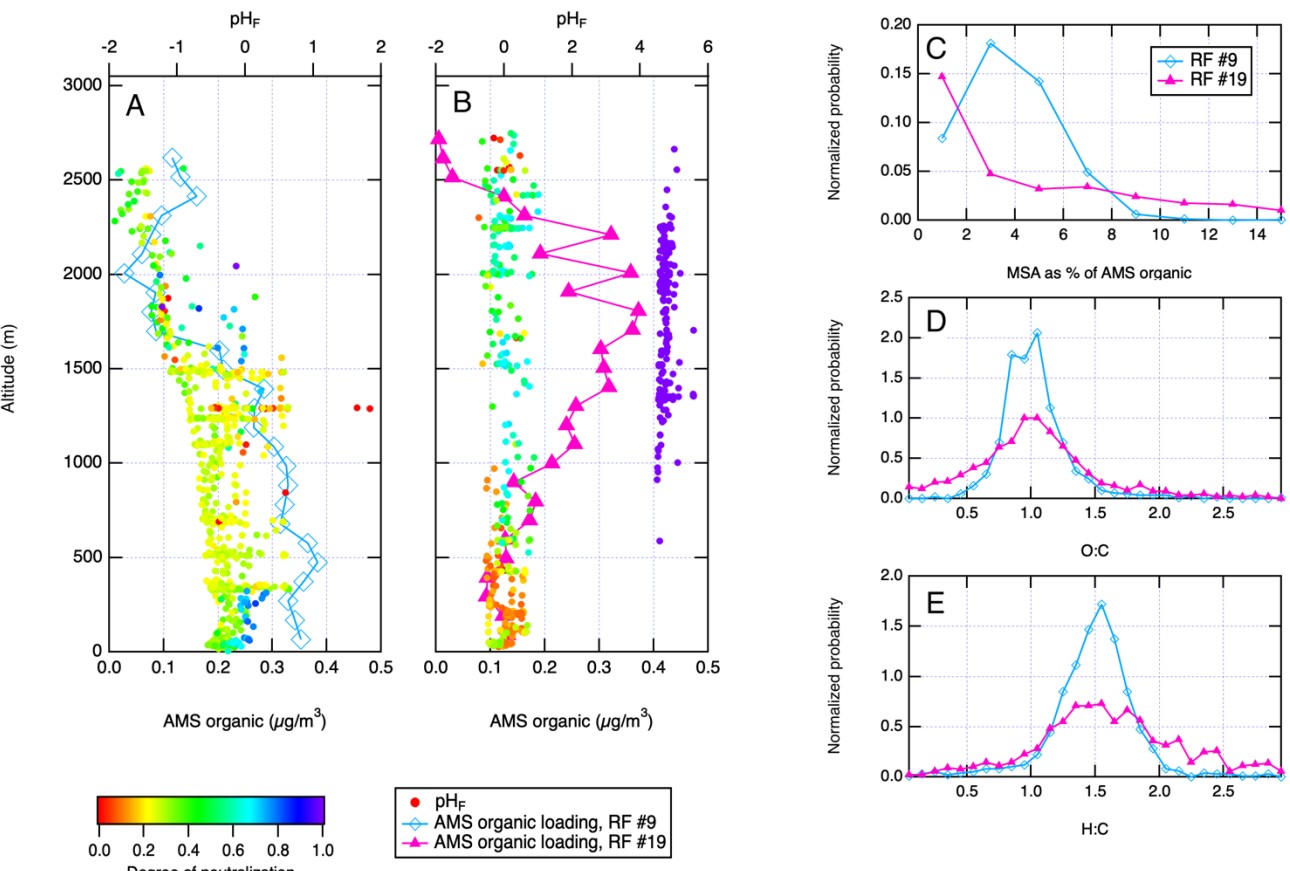

**Figure 6.** Vertical profiles of AMS organic loading for two research flights representative of different (local vs. remote) transport conditions, overlaid with the calculated $pH_F$, (A) RF #9 and (B) #19. (C) Percentage of total AMS organic signal accounted for by MSA during RF #9 and #19, (D) AMS O:C ratios during RF #9 and #19, (E) AMS H:C ratios during RF #9 and #19.





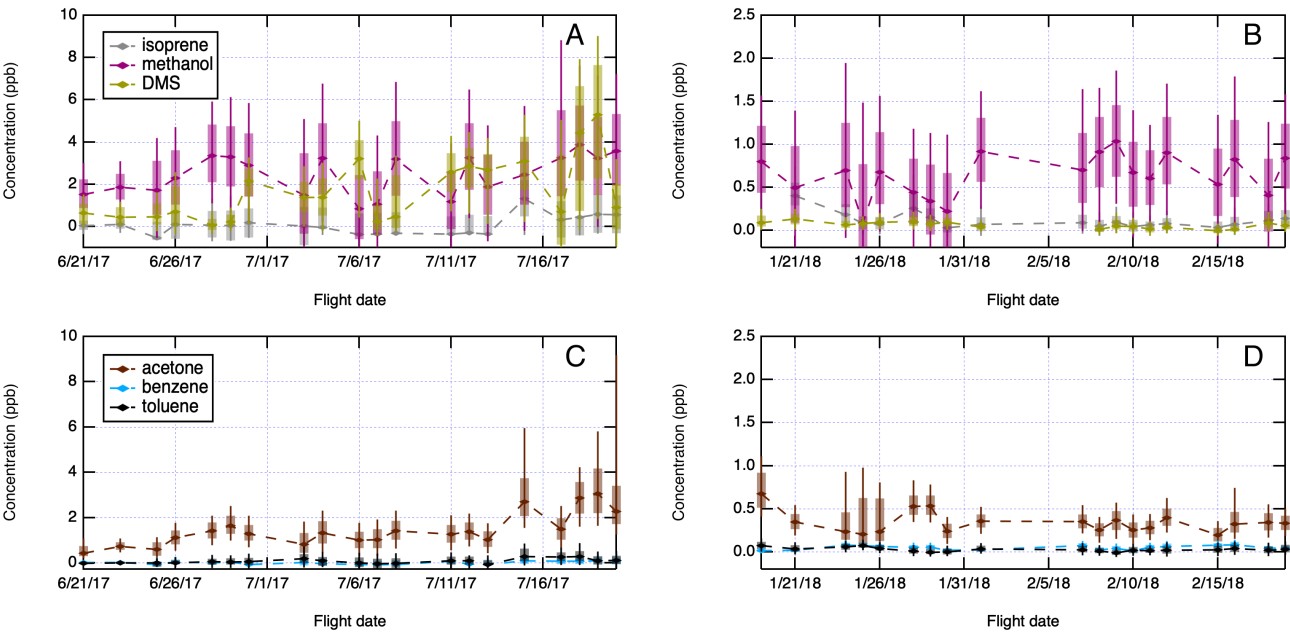


**Figure 7.** Summary of trace gas chemistry measurements acquired with the PTR-MS aboard the G-1 aircraft during ACE-ENA. Measurements at altitudes from 1000 m to 3000 m. Boxes are centered on daily medians, box boundaries extend between 25th and 75th percentile, and whiskers extend between 10th and 90th percentile. (A) and (C) IOP 1 measurements. (B) and (D) IOP 2 measurements.






**Figure 8.** Vertical profiles of AMS aerosol chemistry measurements acquired during spiral profiles shown in Supplementary Figure S1. Representative error bars (1σ standard deviations) are shown on one day only for clarity. (A) Organic, IOP 1. (B) Organic, IOP 2. (C) Sulfate, IOP 1. (D) Sulfate, IOP 2. (E) Ammonium, IOP 1. (F) Ammonium, IOP 2. (G) Nitrate, IOP 1. (H) Nitrate, IOP 2.





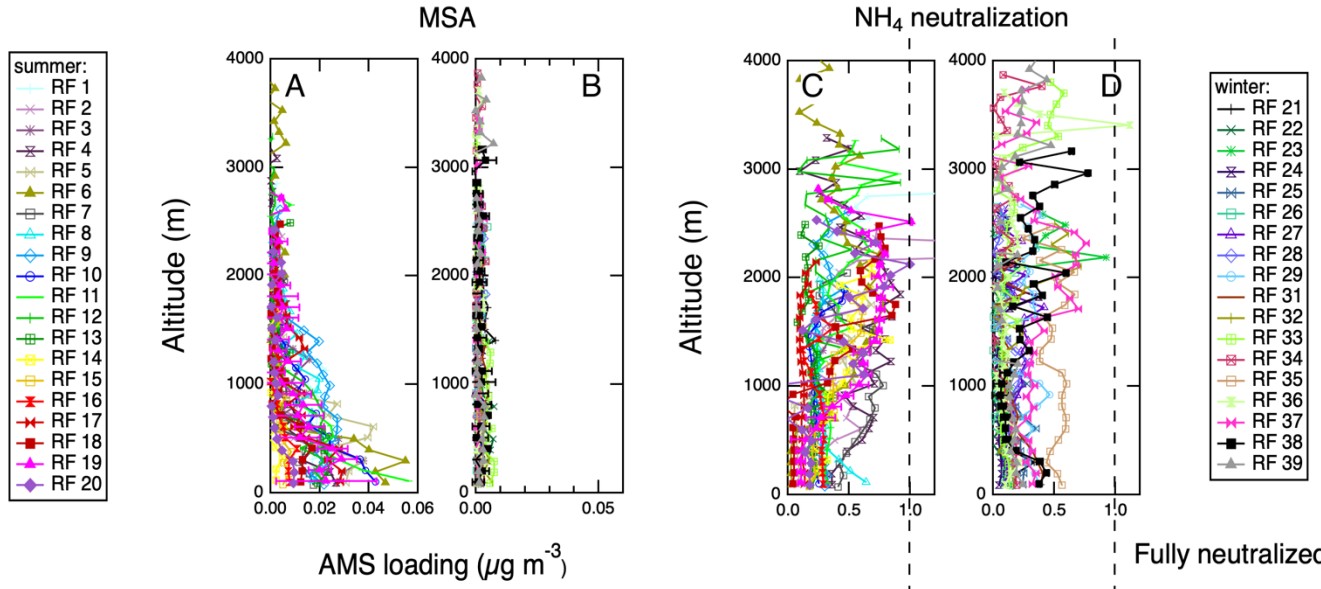

**Figure 9.** Vertical profiles of AMS-derived MSA and AMS-derived degree of NH4 neutralization (a proxy for aerosol acidity) acquired during spiral profiles shown in Supplementary Figure S1. Representative error bars (1σ standard deviations) are shown on one day only for clarity. (A) MSA, IOP 1. (B) MSA, IOP 2. (C) NH4 neutralization, IOP 1. (D) NH4 naturalization, IOP 2.





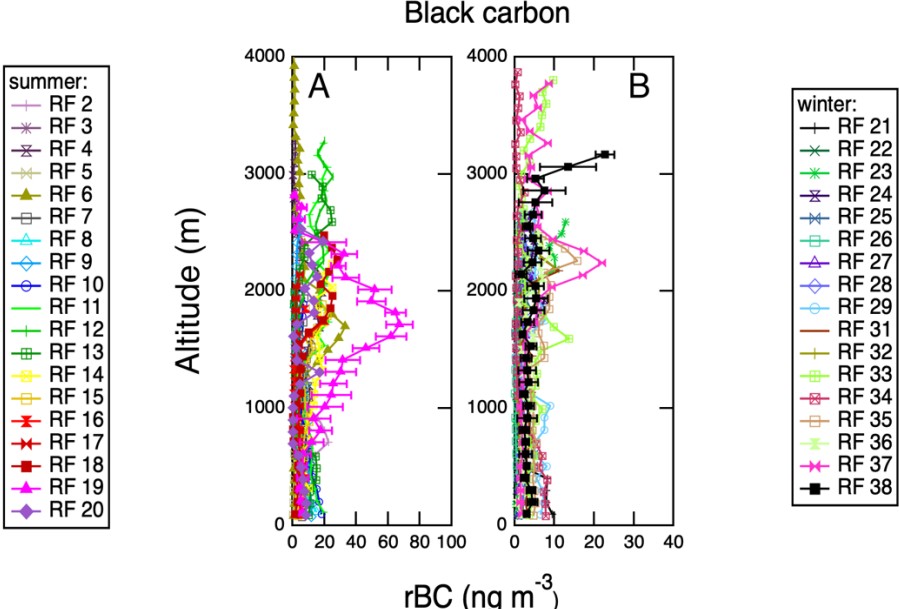

**Figure 10.** Vertical profiles of refractory black carbon (rBC) acquired with SP2 during spiral profiles shown in Supplementary

Figure S1. Representative error bars (1σ standard deviations) are shown on one day only for clarity. (A) IOP 1. (B) IOP 2.





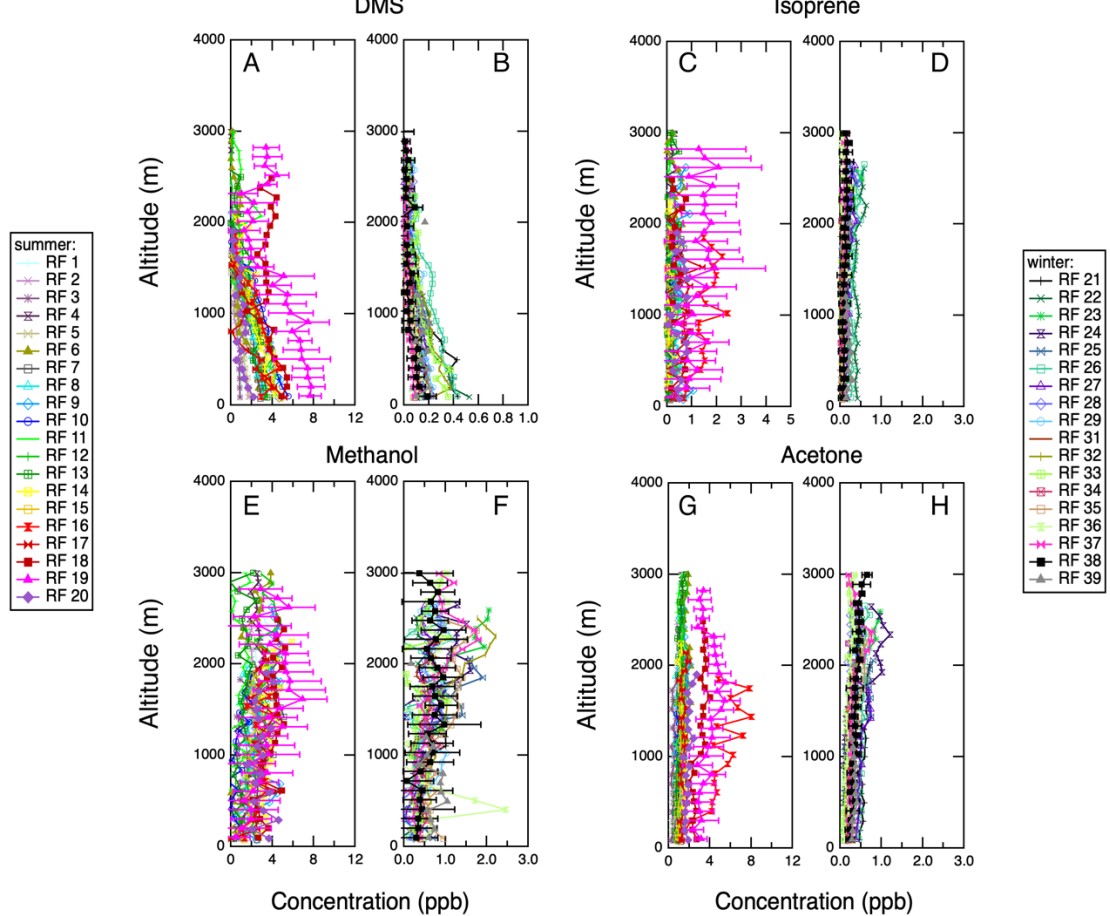

**Figure 11.** Vertical profiles of PTR-MS trace gas chemistry measurements acquired during spiral profiles shown in Supplementary Figure S1. Representative error bars (1σ standard deviations) are shown on one day only for clarity. (A) Dimethyl sulfide, IOP 1. (B) Dimethyl sulfide, IOP 2. (C) Isoprene, IOP 1. (D) Isoprene, IOP 2. (E) Methanol, IOP 1. (F) Methanol, IOP 2. (G) Acetone, IOP 1. (H) Acetone, IOP 2.





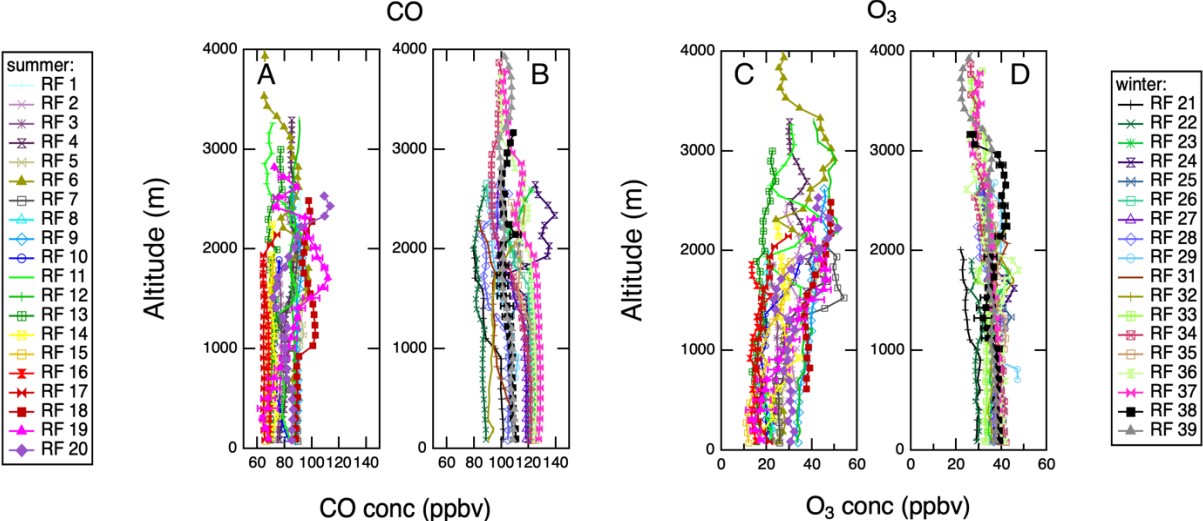

**Figure 12.** Vertical profiles of carbon monoxide and ozone measurements acquired during spiral profiles shown in Supplementary Figure S1. Representative error bars (1σ standard deviations) are shown on one day only for clarity. (A) CO, IOP 1. (B) CO, IOP 2. (C) Ozone, IOP 1. (D) Ozone, IOP 2.





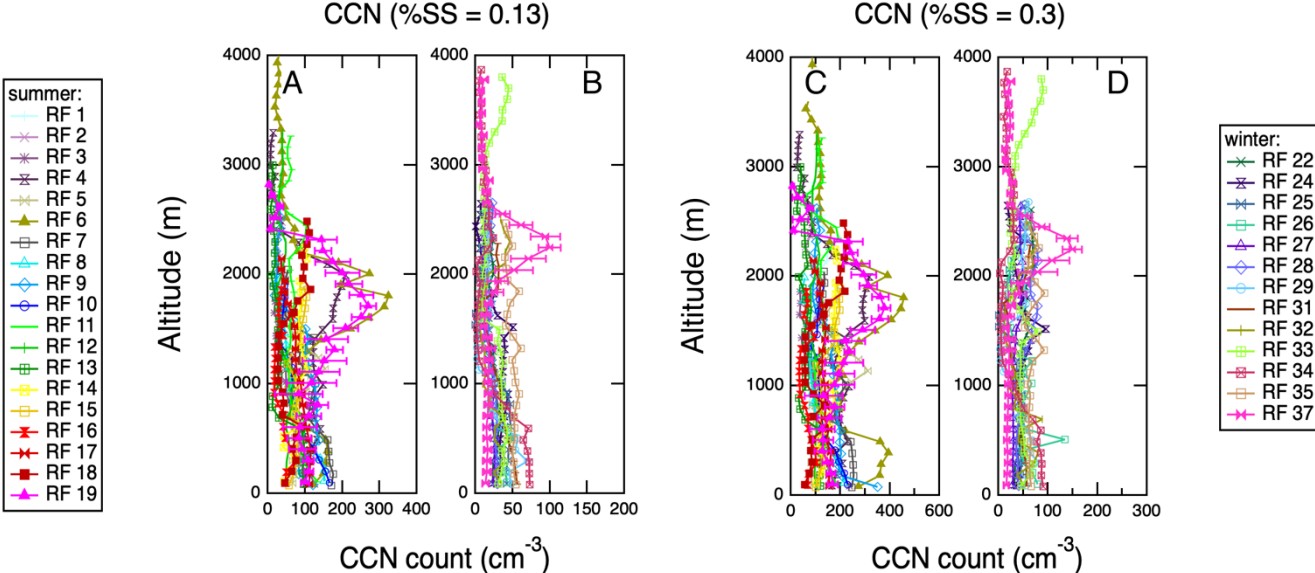

**Figure 13.** Vertical profiles of CCN concentration measurements at two supersaturations (0.13% and 0.3%) acquired during spiral profiles shown in Supplementary Figure S2. Representative error bars (1σ standard deviations) are shown on one day only for clarity. (A) CCN concentration at 0.13% supersaturation, IOP 1. (B) CCN concentration at 0.13% supersaturation, IOP 2. (C) CCN concentration at 0.3% supersaturation, IOP 1. (D) CCN concentration at 0.3% supersaturation, IOP 2.





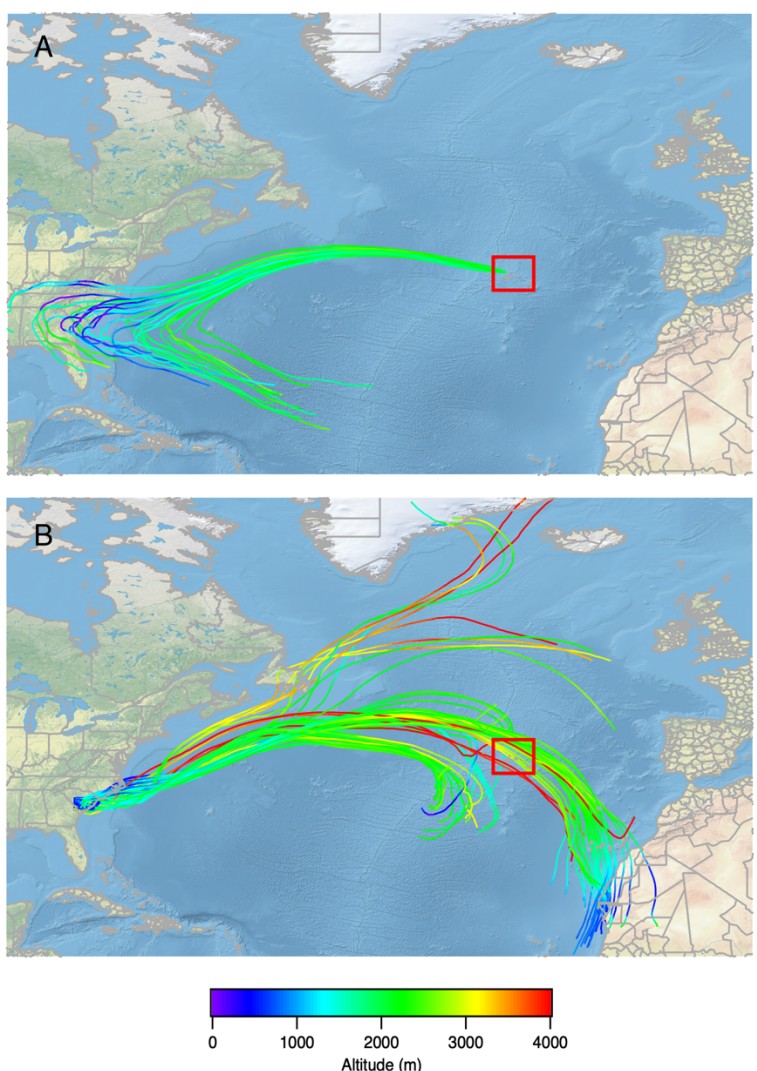


**Figure 14:** HYSPLIT trajectory analysis for the case study of RF #19 (July 19, 2017). (A) A 27-member ensemble of 14-day back-trajectories was started from the first vertical profile location (39.172 N, 28.379 W) at 2000 m altitude. Each member of the trajectory ensemble is calculated by offsetting the meteorological data by a fixed grid factor (one grid meteorological grid point in the horizontal and 0.01 sigma units in the vertical). (B) A matrix of 80 10-day forward-trajectories was started from

an evenly spaced grid bounded by (34 N, 81.75 W), (34 N, 79.5 W), (32.6 N, 81.75W) and (32.6 N, 79.5W) at 500 m altitude. In both cases, GDAS 0.5 degree meteorology and isentropic vertical motion were used. The red box indicates the location of the Azores. The map was created using public domain map data on Natural Earth (naturalearthdata.com) and the GSHHG Database (ngdc.noaa.gov/mgg/shorelines/).