# Peer review of "Aircraft measurements of aerosol and trace gas chemistry in the Eastern North Atlantic"

_Atmospheric Chemistry and Physics, 2020_

## Referee Comment (RC1) · Anonymous Referee #1 · 1 Dec 2020

The paper presents results from two campaigns of aircraft measurements over the eastern North Atlantic based out of the Azores. Vertical profiles of non-refractory aerosol composition, CCN, and trace gas composition are reported and found to confirm previously reported results for the North Atlantic (e.g., both local and long-range transported emissions contribute to the aerosol population in this region). As indicated in the comments below, the addition of references to previously reported results would enhance the paper. The profiles of gas phase species are a strong contribution since there are few similar measurements over oceans. The paper should be published after the comments below are addressed.

Lines 62 – 63: This statement ("...this view (the CLAW hypothesis) has been debated as primary sea salt aerosols....have been hypothesized to be a more robust source

of CCN than DMS-derived aerosols") is an inaccurate and oversimplified summary of Quinn and Bates (2011). There are at least three factors that prevent DMS-derived sulfate from being involved in a climate feedback mechanism – 1) there are additional sources of CCN in the MBL (e.g., sea spray), 2) nucleation of new particles from DMS sulfur occurs in the free troposphere prohibiting local feebacks, and 3) the connection between increased CCN and changes in aerosol-cloud interactions is more complicated than depicted in Charlson et al. (1987).

Lines 109 – 111: NAAMES took place in the western North Atlantic region – west of 30W.

Lines 123 – 124: Perhaps rephrase this statement as "A unique feature of the ACE-ENA aircraft deployments is the seasonally-resolved measurement IN THE EASTERN NORTH ATLANTIC. . ." as NAAMES also had seasonally resolved aircraft flights but in the western NA.

Section 3.1.1 and throughout: To avoid confusion, it should be pointed out that total particulate sulfate measured with the AMS is non-sea salt sulfate.

Lines 190 – 209: It would be interesting to add a comparison with NAAMES seasonal sulfate values. See Saliba et al., JGR, 125, doi: 10.1029/2020JD033145 (2020) and Sanchez et al. Sci. Rep. 8, 3235, doi: 10.1038/s41598-018-21590-9 (2018).

Lines 210 – 220: Seasonal concentrations of DMS and MSA from NAAMES could be compared to the values measured here. See Quinn et al., JGR, 124, 14240 – 14261, 2019. In addition, the MSA to non-sea salt sulfate ratio measured here during the summer (<10%) should be compared to previously reported ratios in remote marine regions during the spring/summer.

Line 229 – 230: This result (MSA does not account for the majority of the particulate sulfate mass in the MBL) is not new and should be noted as such by providing appropriate references.

Line 236: Figure 9 is mentioned before Figures 5 to 8.

Lines 256 – 257: What was used to designate RF #9 and #19 as having lower and higher influence from long range transport?

Lines 275 – 276: Are measured levels of isoprene and monoterpene consistent with this statement, i.e., are they large enough to provide the third source of organic aerosol at ENA? Based on Figure 11, there is no significant surface (marine) source of isoprene even in summer.

Lines 278 – 286: The result that methanol concentrations are larger than DMS in the summertime should be provided with a caveat that reflects the results shown in Figure 11, i.e., methanol concentrations are higher aloft ($\sim$2000 m) while DMS concentrations are lower near the surface.

Lines 297 – 298: The winter surface source of sulfate appears to only be significant in RF 34. What would a winter source of sulfate be? DMS concentrations should be quite low. Figure 11b indicates that DMS surface concentrations were low during RF 34.

Line 327: 0.1 or 0.13%S?

Figures S3 and S4: There appears to be two different populations showing up in Figure S3d and S4d. What is causing this split response between CCN and AMS SO4? It looks like it could be partially responsible for the low r^2 values.

Lines 345 – 349 and Fig. S5: It is really difficult to see that winter concentrations of chlorophyll are higher than summer concentrations in this figure.

---

## Referee Comment (RC2) · Steven Howell (Referee) · 11 Dec 2020

**Comments on "Aircraft measurements of aerosol and trace gas chemistry in the Eastern North Atlantic" by Zawadowicz et al.**

Steven Howell

December 10, 2020

**ACP criteria**

**Does the paper address relevant scientific questions within the scope of ACP?**
Yes

**Does the paper present novel concepts, ideas, tools, or data?** Yes, primarily the data.

**Are substantial conclusions reached?** This is weak, as discussed below. There's little in the way of new conclusions.

**Are the scientific methods and assumptions valid and clearly outlined?** Pretty much.

**Are the results sufficient to support the interpretations and conclusions?** Yes.

**Is the description of experiments and calculations sufficiently complete and precise to allow the** yes, insofar as it's possible to reproduce an experiment where the weather matters.

**Do the authors give proper credit to related work and clearly indicate their own new/original** Yes.

**Does the title clearly reflect the contents of the paper?** Yes.

**Does the abstract provide a concise and complete summary?** Yes.

**Is the overall presentation well structured and clear?** Pretty good.

**Is the language fluent and precise?** Generally yes.

**Are mathematical formulae, symbols, abbreviations, and units correctly defined and used?**
Generally yes, though formulae are not formatted according to Copernicus rules.

**Should any parts of the paper (text, formulae, figures, tables) be clarified, reduced, combined,** Some of the plots aren't really easy to read.

**Are the number and quality of references appropriate?** That's discussed below.

**Is the amount and quality of supplementary material appropriate?** Yes.

**Overall comments**

This paper was a bit frustrating to read. The experimental work seems to have been done well, and it appears to be a valuable dataset for future work, but there are three fundamental issues that are given short shrift here and really ought to be improved before the paper is accepted: the literature review is woefully inadequate; the meteorological context is largely omitted, and much more could have been done with the discussion.

**Literature review**

The authors seem to have spent very little effort seeking prior data about aerosol chemistry projects near their study area, even though Azores and surrounding ocean have been a pretty common site for field projects. They did not have the time resolution, CIMS or CCN, but the sulfur chemistry in the central North Atlantic has been studied at least as far back as 1987 (Galloway, Penner, et al. 1992; Galloway, Keene, et al. 1990; Pszenny et al. 1990) and the ASTEX/MAGE project, employing aircraft, ground stations, and a ship, was based in the Azores in 1992 (Blomquist, Bandy, and Thornton 1996; Huebert et al. 1996; Zhuang and Huebert 1996). I was really puzzled that no mention was made of them. The ACE-2 project (Quinn et al. 2000; Raes et al. 2000) was centered a bit farther south, but seems relevant, as do some oceanographic cruises (e.g. Andreae et al. 2003).

The authors apparently made the rather peculiar decision that only other projects featuring AMS measurements were worth considering, resulting in rather tenuously related comparisons (the Amazon basin, and biomass burning plumes from Southern Africa, for example.) Even then, there are more appropriate comparisons than the ones used. VO-CALS looked at another clean cloud deck (Shank et al. 2012; Wood et al. 2011). ATOM passed through your study area 4 times with a huge payload, including AMS and CIMS (Wofsy et al. 2018).

I cannot fathom why there is no mention of any of the many studies based at Mace Head, on the west coast of Ireland (e.g. Dall'Osto et al. 2010; Ovadnevaite et al. 2014).

**Meterolological context**

The meteorological setting is woefully incomplete. What was typical cloud height? Was there usually an extensive stratocumulus deck? What altitude? Was the boundary layer decoupled? Was there a strong inversion? Was there a buffer layer? Was there drizzle that

might be scavenging aerosol? Why were the altitude ranges $0\,\text{m}$ to $1000\,\text{m}$ and $1000\,\text{m}$ to $3000\,\text{m}$ chosen? I presume it had to do with boundary layer depth, but that should be explicit. Using fixed altitude ranges is convenient, but does it really reflect the structure of the atmosphere? It is often best to group data into mixed layer, cloud layer, buffer layer (if present), and free troposphere.

Are there estimates of entrainment velocity during the experiment? That would constrain how fast the MBL is diluted with FT air and thus give a lower bound on MBL sources of aerosol. Or was the meteorological situation too complex for such approximations to make sense?

**Discussion**

Since the literature survey was so minimal, I suppose it was inevitable that there would be a lack of direct comparison with earlier data from the region. But it would be interesting to know whether things have changed. Of particular interest is the fact that earlier experiments found much more $MS^-$. Is that a real change, or is it possibly that much of the $MS^-$ was on particles too large for the AMS? Were the $MS^-$:DMS or $MS^-$:$SO_4^{2-}$ ratios different? Is the aerosol sufficiently acidic to drive off $HNO_3$? (I strongly suspect that's why you saw so little $NO_3^-$; it was displaced to seasalt particles too big for the AMS to detect.)

I was also a bit surprised to see no estimates of how much of the aerosol in the boundary layer could have come from the free troposphere. Without level legs at the top of the MBL, you obviously cannot do a flux study, but you have tracers of long-distance transport that are immune to precipitation scavenging (CO, benzene, toluene) and at least 1 that is not (BC). Can you use those to put some bounds on how much of the $SO_4^{-2}$ and organic aerosol came from above? Would that bring the $MS^-$:$SO_4^{2-}$ in line with other work?

The dramatic split in Fig. 6b in $pH_F$ is interesting. Since Table S1 says there were 5 profiles on 19 July, it seems likely that some profiles were in the putative fire plume while others were not. It might be worth trying multiple back trajectories on each of the profiles to establish a pattern like that seen in Clarke et al. (2013), where the ensemble of trajectories and their correlation with aerosol properties lent credibility to the trajectories over rather long distances.

Those early experiments had far less data, but did more with it, examining budgets and processes, not just using back trajectories to conclude that sometimes continental aerosol was present and sometimes it was really clean. You have more, better quality data, and it's integrated with a larger instrument package, so you ought to be able to do more.

**More specific comments**

**line 21** "fully" is meaningless. It suggests that all possibly relevant instrumentation was aboard, and that's not possible even on much bigger aircraft.

**line 26** average submicrometer non-refractory aerosol mass

**line 33** "1 % of the sulfate and no more than 3 % of the total aerosol" makes no sense. You presumably meant no more than 3 % of the submicron organic aerosol.

**line 140** Was there a typical altitude range of the spirals? "Through the atmosphere" is vague (and strictly speaking, incorrect). Was there a minimum altitude span criterion for inclusion in Table S1? Were there any criteria for where to do the spirals?

**line 145** Was the standard aerodynamic lens used? What temperature was the vaporizer?

**line 155** Has the inlet efficiency for this inlet been characterized?

**line 156** "switched ... based on cloud cover" seems unlikely. Surely switching was done based on whether the plane was actually in cloud.

**lines 166–167** I'm a bit surprised at all the English (rather than SI) units. I guess that's up to the journal editors. In addition, the OD of the tubing is irrelevant; it's the ID that matters.

**line 186** The low supersaturation is presented inconsistently. Here and in Fig. S3 it is 0.1 %, while in Fig. 13 it is 0.13 %.

**line 173** I don't see how elevated DMS background in the summer IOP necessarily biases DMS measurements high. If it is indeed an isobaric interference (any idea what would do that?) then one would eliminate the overestimate by subtracting the background. If it was incomplete destruction of the DMS, then subtracting the background would produce an underestimate. If the interfering species was partly destroyed by the catalyst, then yes you would have an overestimate of DMS, but you could only claim that the patterns you see in DMS are accurate if the interfering species concentration was fairly constant.

**line 182** "more closely mimic" is incomplete. More closely than what?

**line 229–230** Has anyone ever claimed that MSA accounted for the majority of particulate sulfate mass?

**line 235** The equation is only true if those species are the only acids and bases present in the aerosol. There are organic acids like oxalic acid and MSA. They are probably negligible here, but that ought to be noted.

**lines 242–245** Be more explicit about the use of the thermal denuder. Was there very little submicron seasalt (as is likely)? As determined by heating to what temperature? It's true that the AMS doesn't see coarse particles, so whether they are externally mixed seems irrelevant. Is there a claim here that there was little volatile material on coarse particles so the AMS wasn't actually missing significant $MS^-$ and $NO_3^-$?

**line 250** It's not wrong, but a bit odd to use a moving average with an even number of points, meaning that the time represented by the average is between the times of two data points, but not right on them. I suppose it doesn't matter here since there aren't any comparisons here that depend on close synchronization.

**line 275** Acid-catalyzed reactive uptake of organic vapors is an interesting idea. Any citations for it? Could SOA production via that mechanism be fast enough to account for the extra organic aerosol in the MBL?

**line 371** (This is a hobbyhorse of mine.) Airmasses do not have origins! The air always came from somewhere earlier and has traces of that left in it. If there was near-total scavenging event or a large influx of pollution that dwarfs whatever was present, then one could claim there is an origin of the characteristics of the air mass. Ascribing an origin to an air parcel in the MBL is particularly absurd, since there is almost always entrainment mixing going on meaning that much of the air was in the FT within the last few days. Simple back trajectories are not really capable of conveying that.

**line 402** Surely you men "summer and winter" rather than "summer and fall".

**Fig. 4** It's jarring that the pie charts for the organic fraction of the aerosol are larger than the the total aerosol.

**Fig. 6** This is an interesting figure, but the caption isn't as clear as it ought to be. It appears that in panels A and B, the organic loading is averaged into $100\,\mathrm{m}$ bins, while the $\mathrm{pH_F}$ is for individual ($10\,\mathrm{s}$?) averages. The $\mathrm{pH_F}$ data in panel B has a remarkable split between nearly neutralized and very acidic aerosol, as though some of the 5 spirals that day were in the pollution plume while the others were not. Of course the averaged organic loading cannot show that. Would it be worth plotting the spirals separately, or grouped as plume vs. non-plume? Were panels C, D, and E for the entire vertical profile? What does "normalized" mean here? Same area under the curve? It didn't happen often, but it appears from panel C that $\mathrm{MS^-}$ was sometimes $10\,\%$ to $15\,\%$ of the $\mathrm{SO_4}^{2-}$. Is that real?

**Fig. 14** While the maps are quite pretty, there is a lot of information, such as bathymetry, that is unimportant to the paper. I don't actually object much to that even though it is best practice to exclude irrelevant material from graphics. However, including the political divisions within countries is clearly excessive.

**Equations in the supplement** I assume the Copernicus editors will help you figure out what should be italicized and what should not be.

**CE of MSA in the supplement** This is a misinterpretation of Middlebrook et al. (2012). It's not the pH that matters–it's whether the aerosol is liquid or solid. That said, since MSA salts are solid and MSA itself is liquid (much like $H_2SO_4$ and salts thereof),

the CEs you propose are reasonable in the lab. I'm not sure what you're doing with the field data–the particles are presumably internal mixtures with only small contributions from $MS^-$. In that case, it's the presence of liquid $H_2SO_4$ that will determine CE for the entire aerosol.

**PIKA vs Squirrel** Did you get $I_{CH_3SO_2^+}$, $I_{CH_2SO_2^+}$ and $I_{CH_4SO_3^+}$ from PIKA or did you use unit mass data from Squirrel? I guess you did it with PIKA, which would make sense, particularly in the field, where other species would be present at those unit masses. To look for interferences, it might be useful to plot $I_{CH_4SO_3^+}$ vs $I_{CH_3SO_2^+}$ and $I_{CH_2SO_2^+}$ vs $I_{CH_4SO_3^+}$ to see whether you have the same fragmentation pattern in the field as you had in the lab.

**Table S1** Including 4 digits after the decimal point for latitude and longitude specifies the point to within $11\,\mathrm{m}$. Seems excessive. Might be useful to add an altitude range, unless that was constant (in which case I'd like to see that somewhere).

**Fig. S3** The $y$ axes on panels E and F are labeled $SO_4$ rather than $NH_4$.

**References**

Andreae, M., T. Andreae, D. Meyerdierks, and C. Thiel (2003). "Marine sulfur cycling and the atmospheric aerosol over the springtime North Atlantic". *Chemosphere* 52.8, pp. 1321–1343. DOI: https://doi.org/10.1016/S0045-6535(03)00366-7.

Blomquist, B., A. Bandy, and D. Thornton (1996). "Sulfur gas measurements in the eastern North Atlantic Ocean during the Atlantic Stratocumulus Transition Experiment/Marine Aerosol and Gas Exchange". *J. Geophys. Res.–Atmos.* 101.D2, pp. 4377–4392.

Clarke, A. D., S. Freitag, R. M. C. Simpson, J. G. Hudson, S. G. Howell, V. L. Brekhovskikh, T. Campos, V. N. Kapustin, and J. Zhou (2013). "Free troposphere as a major source of CCN for the Equatorial Pacific boundary layer: long-range transport and teleconnections". *Atmos. Chem. Phys.* 13.15, pp. 7511–7529. DOI: 10.5194/acp-13-7511-2013.

Dall'Osto, M., D. Ceburnis, G. Martucci, J. Bialek, R. Dupuy, S. G. Jennings, H. Berresheim, J. Wenger, R. Healy, M. C. Facchini, M. Rinaldi, L. Giulianelli, E. Finessi, D. Worsnop, M. Ehn, J. Mikkilä, M. Kulmala, and C. D. O'Dowd (2010). "Aerosol properties associated with air masses arriving into the North East Atlantic during the 2008 Mace Head EUCAARI intensive observing period: an overview". *Atmos. Chem. Phys.* 10.17, pp. 8413–8435. DOI: 10.5194/acp-10-8413-2010.

Galloway, J. N., J. E. Penner, C. S. Atherton, et al. (1992). "Sulfur and Nitrogen and Oxidant Levels in the North Atlantic Ocean's Atmosphere: A Synthesis of Field and Modeling Results". *Global Biogeochem. Cy.* 6, pp. 77–100.

Galloway, J. N., W. C. Keene, A. A. P. Pszenny, D. M. Whelpdale, H. Sievering, J. T. Merrill, and J. F. Boatman (1990). "Sulfur in the western North Atlantic Ocean atmosphere: results from a summer 1988 ship/aircraft experiment". *Global Biogeochem. Cy.* 4.4, pp. 349–65.

Huebert, B. J., L. Zhuang, S. Howell, K. Noone, and B. Noone (1996). "Sulfate, Nitrate, Methanesulfonate, Chloride, Ammonium, and Sodium Measurements from Ship, Island, and Aircraft during ASTEX/MAGE". *J. Geophys. Res.–Atmos.* 101.D2, pp. 4413–4423.

Ovadnevaite, J., D. Ceburnis, S. Leinert, M. Dall'Osto, M. Canagaratna, S. O'Doherty, H. Berresheim, and C. O'Dowd (2014). "Submicron NE Atlantic marine aerosol chemical composition and abundance: Seasonal trends and air mass categorization". *J. Geophys. Res.–Atmos.* 119.20, pp. 11, 850–11, 863. DOI: `10.1002/2013JD021330`.

Pszenny, A. A. P., G. R. Harvey, C. J. Brown, R. F. Lang, W. C. Keene, J. N. Galloway, and J. T. Merrill (1990). "Measurements of dimethyl sulfide oxidation products in the summertime north Atlantic marine boundary layer". *Global Biogeochem. Cy.* 4.4. CASE/WATOX results for DMSO, DMSO2, and sulfate aerosols in mbl. Estimate a .3 enhancement in total S deposition due to anthro. effects. DMSO and DMSO2 more important than MSA in deposition to sea., pp. 367–379.

Quinn, P. K., T. S. Bates, D. J. Coffman, T. L. Miller, J. E. Johnson, D. S. Covert, J.-P. Putaud, C. Neusüß, and T. Novakov (2000). "A comparison of aerosol chemical and optical properties from the 1st and 2nd Aerosol Characterization Experiments". *Tellus B* 52.2, pp. 239–257. DOI: `10.3402/tellusb.v52i2.16103`.

Raes, F., T. Bates, F. McGovern, and M. V. Liedekerke (2000). "The 2nd Aerosol Characterization Experiment (ACE-2): general overview and main results". *Tellus B* 52.2, pp. 111–125. DOI: `10.3402/tellusb.v52i2.16088`.

Shank, L. M., S. Howell, A. D. Clarke, S. Freitag, V. Brekhovskikh, V. Kapustin, C. McNaughton, T. Campos, and R. Wood (2012). "Organic matter and non-refractory aerosol over the remote Southeast Pacific: oceanic and combustion sources". *Atmospheric Chemistry and Physics* 12.1, pp. 557–576. DOI: `10.5194/acp-12-557-2012`.

Wofsy, S., S. Afshar, H. Allen, et al. (2018). *ATom: Merged Atmospheric Chemistry, Trace Gases, and Aerosols.* en. DOI: `10.3334/ORNLDAAC/1581`.

Wood, R., C. R. Mechoso, C. S. Bretherton, et al. (2011). "The VAMOS Ocean-Cloud-Atmosphere-Land Study Regional Experiment (VOCALS-REx): goals, platforms, and field operations". *Atmos. Chem. Phys.* 11.2, pp. 627–654. DOI: `10.5194/acp-11-627-2011`.

Zhuang, L. and B. J. Huebert (1996). "A Langrangian Analysis of the Total Ammonia Budget during ASTEX/MAGE". *J. Geophys. Res.–Atmos.* 101.D2, pp. 4341–4350.

---

## Author Comment (AC1) · 25 Feb 2021

We thank the reviewer for their comments. Below we reproduce the comments in blue and provide our discussion in black.

Lines 62 – 63: This statement (". . .this view (the CLAW hypothesis) has been debated as primary sea salt aerosols. . ..have been hypothesized to be a more robust source of CCN than DMS-derived aerosols") is an inaccurate and oversimplified summary of Quinn and Bates (2011). There are at least three factors that prevent DMS-derived sulfate from being involved in a climate feedback mechanism – 1) there are additional sources of CCN in the MBL (e.g., sea spray), 2) nucleation of new particles from DMS sulfur occurs in the free troposphere prohibiting local feebacks, and 3) the connection between increased CCN and changes in aerosol-cloud interactions is more complicated than depicted in Charlson et al. (1987).

We agree with the reviewer that the original statement was oversimplified. We revised it to read, "More recently, it has been shown that there are factors, such as additional sea salt sources of CCN in the MBL or complex interactions between CCN and aerosol-cloud interactions, that prevent DMS-derived sulfate from being directly involved in a climate feedback mechanism (Quinn and Bates, 2011).

Lines 109 – 111: NAAMES took place in the western North Atlantic region – west of 30W.

This has been rewritten as "Other recent field measurements focusing on the aerosol chemistry of the North Atlatic region include the NASA North Atlantic Aerosols and Marine Ecosystems Study (NAAMES), which included both aircraft and shipborne observations focused on marine biological productivity of the Western North Atlantic (Behrenfeld et al., 2019)…"

Lines 123 – 124: Perhaps rephrase this statement as "A unique feature of the ACEENA aircraft deployments is the seasonally-resolved measurement IN THE EASTERN NORTH ATLANTIC. . ." as NAAMES also had seasonally resolved aircraft flights but in the western NA.

This has been revised according to reviewer's suggestion.

Section 3.1.1 and throughout: To avoid confusion, it should be pointed out that total particulate sulfate measured with the AMS is non-sea salt sulfate.

We have added the non-sea salt qualifier to sulfate.

Lines 190 – 209: It would be interesting to add a comparison with NAAMES seasonal sulfate values. See Saliba et al., JGR, 125, doi: 10.1029/2020JD033145 (2020) and Sanchez et al. Sci. Rep. 8, 3235, doi: 10.1038/s41598-018-21590-9 (2018).

We added the comparison, "The NAAMES cruises in the Western North Atlantic provide another point of comparison: in the winter, the loadings of organic, non-sea salt sulfate, ammonium and nitrate were 0.14/0.56, 0.15/0.48, <0.01/0.01 and 0.01/0.03, respectively, and in the late spring they were 0.61/1.62, 0.44/0.64, <0.01/0.13 and 0.02/0.14, respectively, for marine/continental airmass origin (Saliba et al., 2020)."

Lines 210 – 220: Seasonal concentrations of DMS and MSA from NAAMES could be compared to the values measured here. See Quinn et al., JGR, 124, 14240 – 14261, 2019. In addition, the MSA to non-sea salt sulfate ratio measured here during the summer (<10%) should be compared to previously reported ratios in remote marine regions during the spring/summer.

We now compare our reported MSA measurements to NAAMES measurements reported in Quinn, et al. (2019), "Similarly, during the NAAMES cruises, MSA concentrations measured with ion chromatography were reported as 0.07 µg m$^{-3}$ in the late spring, and 0.01 µg m$^{-3}$ in both March and September (Quinn et al., 2019; Saliba et al., 2020).".

We decided against comparing our DMS values to NAAMES, as our values potentially suffer from interferences in the summer, as explained in the paper, and we don't want to be misleading about the accuracy of our DMS measurements.

We also added the following discussion of MSA to non-sea sulfate ratio, "The MSA to non-sea salt sulfate ratio (MSA:SO$_4$) measured during ACE-ENA in the summer was 0.02 on average in the MBL (<1000 m), which is lower than historical estimates of the ratio. For example, Pszenny et al. (1990) reports the ratio as 0.05 in the North Atlantic in August-September, Berresheim et al. (1991) reports 0.033 in Western North Atlantic in September, Savoie et al. (2002) reports 0.06 in Bermuda in September and 0.05 in Mace Head in August. Huebert et al. (1996) found 0.07 in marine air masses and 0.02 in continental air masses in June in the Azores. The measurements reported in these earlier studies are based on analysis of filter samples, and they may not be directly comparable to AMS measurements reported here."

Line 229 – 230: This result (MSA does not account for the majority of the particulate sulfate mass in the MBL) is not new and should be noted as such by providing appropriate references.

This sentence was misleading and was removed.

Discussion of Figure 9 was moved to Section 3.2

Lines 256 – 257: What was used to designate RF #9 and #19 as having lower and higher influence from long range transport?

The classification of those cases was done on the basis of vertical profiles, RF #19 has a clear layer of black carbon, sulfate and ammonium above the boundary layer (1000 – 3000 m), as seen in Figures 8 and 10. This is also correlated with increased methanol and acetone (Figure 11), all markers for continental transport. RF #9 looks very clean in comparison with very little black carbon, ammonium or trace gases other than DMS. We added this explanation, "To further test this, two representative research flights, RF #9 and #19 were selected to represent conditions with lower (RF #9) and higher (RF #19) influence from long-range transport on the basis of concentrations of black carbon, ammonium, methanol and acetone in the free troposphere, which are all markers for long-range transport."

Lines 275 – 276: Are measured levels of isoprene and monoterpene consistent with this statement, i.e., are they large enough to provide the third source of organic aerosol at ENA? Based on Figure 11, there is no significant surface (marine) source of isoprene even in summer.

It is correct that we did not measure significant concentrations of isoprene and monoterpenes. However, given that our detection limit for isoprene is approximately 100 ppt (quantified as $3\sigma$ of blank measurements) and that acidic aerosols significantly enhance the SOA production from isoprene and its oxidation products, we can't completely rule out this mechanism. We added the following explanation, "While the isoprene concentrations measured during ACE-ENA were low and close to the detection limit of 0.1 ppb (Table 2) SOA formation from acid-catalyzed IEPOX chemistry has been shown to be significantly more efficient than from non-IEPOX isoprene photochemical mechanisms (Surratt et al., 2010)."

Lines 278 – 286: The result that methanol concentrations are larger than DMS in the summertime should be provided with a caveat that reflects the results shown in Figure 11, i.e., methanol concentrations are higher aloft (~2000 m) while DMS concentrations are lower near the surface.

We added, "Methanol concentrations were higher above the boundary layer (>1000 m), but lower near the ocean surface, while DMS showed the opposite trend of high concentrations at the surface and low concentrations at high altitudes."

Lines 297 – 298: The winter surface source of sulfate appears to only be significant in RF 34. What would a winter source of sulfate be? DMS concentrations should be quite low. Figure 11b indicates that DMS surface concentrations were low during RF 34.

It is unclear what the source of sulfate is in RF #34, in the absence of DMS. Possibilities include remote transport that has mixed downward or an isolated local source, perhaps from a particularly polluting ship transiting through the area. The reviewer is correct to point out that sulfate has a surface source in the summer, and a high altitude source in both seasons. We clarified as follows, "Sulfate aerosol has two distinct sources, a marine source seen below 1000 m in the summer, and a high altitude source above 1000 m present in both seasons."

It's 0.13%. This was fixed.

Figures S3 and S4: There appears to be two different populations showing up in Figure S3d and S4d. What is causing this split response between CCN and AMS SO4? It looks like it could be partially responsible for the low r^2 values.

The reviewer raises an interesting question. Unfortunately, it is difficult to know exactly why the sulfate/CCN plots appear to show different populations while the organic/CCN plots do not. Our hypothesis is that this is related to the aerosol size distribution and mixing state. As described in the manuscript, sulfate has both remote and local sources. New particle formation from DMS-derived sulfate would produce particles that are likely smaller and less CCN-active (due to their

size) on average than particles from long-range transport. In contrast, the organic mass is likely to be largely from remote sources and therefore likely concentrated in larger sized particles. Unfortunately, we do not have size-dependent chemical measurements to test this hypothesis. The figure below, in which the colors indicate different flights, lends credibility to this hypothesis, as it appears that the SO4 vs CCN correlations cluster by flight.

[Figure]

Lines 345 – 349 and Fig. S5: It is really difficult to see that winter concentrations of chlorophyll are higher than summer concentrations in this figure.

Added insets to the figure that show this better.

**References**

Berresheim, H., Andreae, M. O., Iverson, R. L. and Li, S. M.: Seasonal variations of dimethylsulfide emissions and atmospheric sulfur and nitrogen species over the western north Atlantic Ocean, Tellus B: Chemical and Physical Meteorology, 43(5), 353–372, https://doi.org/10.3402/tellusb.v43i5.15410, 1991.

Huebert, B. J., Zhuang, L., Howell, S., Noone, K. and Noone, B.: Sulfate, nitrate, methanesulfonate, chloride, ammonium, and sodium measurements from ship, island, and aircraft during the Atlantic Stratocumulus Transition Experiment/Marine Aerosol Gas Exchange, J. Geophys. Res., 101(D2), 4413–4423, https://doi.org/10.1029/95JD02044, 1996.

Pszenny, A. A. P., Harvey, G. R., Brown, C. J., Lang, R. F., Keene, W. C., Galloway, J. N. and Merrill, J. T.: Measurements of dimethyl sulfide oxidation products in the summertime North Atlantic marine boundary layer, Global Biogeochem. Cycles, 4(4), 367–379, https://doi.org/10.1029/GB004i004p00367, 1990.

Quinn, P. K. and Bates, T. S.: The case against climate regulation via oceanic phytoplankton sulphur emissions, Nature, 480(7375), 51–56, https://doi.org/10.1038/nature10580, 2011.

Quinn, P. K., Bates, T. S., Coffman, D. J., Upchurch, L., Johnson, J. E., Moore, R., Ziemba, L., Bell, T. G., Saltzman, E. S., Graff, J. and Behrenfeld, M. J.: Seasonal Variations in Western North Atlantic Remote Marine Aerosol Properties, J. Geophys. Res. Atmos., 124(24), 14240–14261, https://doi.org/10.1029/2019JD031740, 2019.

Saliba, G., Chen, C., Lewis, S., Russell, L. M., Quinn, P. K., Bates, T. S., Bell, T. G., Lawler, M. J., Saltzman, E. S., Sanchez, K. J., Moore, R., Shook, M., Rivellini, L., Lee, A., Baetge, N., Carlson, C. A. and Behrenfeld, M. J.: Seasonal Differences and Variability of Concentrations, Chemical Composition, and Cloud Condensation Nuclei of Marine Aerosol Over the North Atlantic, J. Geophys. Res. Atmos., 125(19), https://doi.org/10.1029/2020JD033145, 2020.

Savoie, D. L.: Marine biogenic and anthropogenic contributions to non-sea-salt sulfate in the marine boundary layer over the North Atlantic Ocean, J. Geophys. Res., 107(D18), 4356, https://doi.org/10.1029/2001JD000970, 2002.

Surratt, J. D., Chan, A. W. H., Eddingsaas, N. C., Chan, M., Loza, C. L., Kwan, A. J., Hersey, S. P., Flagan, R. C., Wennberg, P. O. and Seinfeld, J. H.: Reactive intermediates revealed in secondary organic aerosol formation from isoprene, Proceedings of the National Academy of Sciences, 107(15), 6640–6645, https://doi.org/10.1073/pnas.0911114107, 2010.

---

## Author Comment (AC2) · 25 Feb 2021

We thank the reviewer for their comments. Below we reproduce the comments in blue and provide our discussion in black.

The authors seem to have spent very little effort seeking prior data about aerosol chemistry projects near their study area, even though Azores and surrounding ocean have been a pretty common site for field projects. They did not have the time resolution, CIMS or CCN, but the sulfur chemistry in the central North Atlantic has been studied at least as far back as 1987 (Galloway, Penner, et al. 1992; Galloway, Keene, et al. 1990; Pszenny et al. 1990) and the ASTEX/MAGE project, employing aircraft, ground stations, and a ship, was based in the Azores in 1992 (Blomquist, Bandy, and Thornton 1996; Huebert et al. 1996; Zhuang and Huebert 1996). I was really puzzled that no mention was made of them. The ACE-2 project (Quinn et al. 2000; Raes et al. 2000) was centered a bit farther south, but seems relevant, as do some oceanographic cruises (e.g. Andreae et al. 2003).

We are now briefly summarizing these previous North Atlantic measurements in the Introduction, "The atmospheric chemistry of North Atlantic region, including the Azores, has been studied in previous oceanographic cruises (Andreae et al., 2003) and aircraft campaigns. Early work on sulfur cycle in the Eastern North Atlantic has been summarized by Galloway et al. (1992). Notable campaigns in the region include the Atlantic Stratocumulus Transition Experiment/Marine Aerosol and Gas Exchange (ASTEX/MAGE) (Blomquist et al., 1996) and the second Aerosol Characterization Experiment (ACE-2) (Raes et al., 2000)."

Some of these papers are also referenced later in the discussion of $MSA:SO_4$ ratios, "The MSA to non-sea salt sulfate ratio ($MSA:SO_4$) measured during ACE-ENA in the summer was 0.02 on average in the MBL (<1000 m), which is lower than historical estimates of the ratio. For example, Pszenny et al. (1990) reports the ratio as 0.05 in the North Atlantic in August-September, Berresheim et al. (1991) reports 0.033 in Western North Atlantic in September, Savoie et al. (2002) reports 0.06 in Bermuda in September and 0.05 in Mace Head in August. Huebert et al. (1996) found 0.07 in marine air masses and 0.02 in continental air masses in June in the Azores. The measurements reported in those earlier studies are based on analysis of filter samples, however, and they may not be directly comparable to AMS measurements reported here."

The authors apparently made the rather peculiar decision that only other projects featuring AMS measurements were worth considering, resulting in rather tenuously related comparisons (the Amazon basin, and biomass burning plumes from Southern Africa, for example.) Even then, there are more appropriate comparisons than the ones used. VOCALS looked at another clean cloud deck (Shank et al. 2012; Wood et al. 2011). ATOM passed through your study area 4 times with a huge payload, including AMS and CIMS (Wofsy et al. 2018).

The comparison with the Amazon basin was made to highlight the cleanliness of the ENA MBL measured in this campaign. The other comparisons made in section 3.1.1 on p. 7 are made on the basis of similarity of marine conditions (Polarstern and NAAMES cruises) and geography (NEAQS flights that occurred on the East coast of the U.S. which would be a source region for aged emissions seen at ENA). We have now added some non-AMS comparisons, cited in the previous reply also centering on the East Atlantic region. We have decided not to compare with

ATom and VOCALS explicitly in order not to clutter section 3.1.1, especially since the location for VOCALS was the Pacific Ocean off the coast of South America. The ATom campaign is also discussed in the literature review portion of the introduction, "Between 2016 and 2017, AMS was also deployed aboard the NASA DC-8 aircraft during the Atmospheric Tomography (ATom) missions in the remote atmosphere, including the North Atlantic region (Hodshire et al., 2019)."

I cannot fathom why there is no mention of any of the many studies based at Mace Head, on the west coast of Ireland (e.g. Dall'Osto et al. 2010; Ovadnevaite et al. 2014).

We added the following, "Of note are also studies on the west coast of Ireland at the Mace Head observatory, which frequently encounter North Atlantic air masses, as described in Dall'Osto et al. (2010) and Ovadnevaite et al. (2014)."

The meteorological setting is woefully incomplete. What was typical cloud height? Was there usually an extensive stratocumulus deck? What altitude? Was the boundary layer decoupled? Was there a strong inversion? Was there a buffer layer? Was there drizzle that might be scavenging aerosol? Why were the altitude ranges 0 m to 1000 m and 1000 m to 3000 m chosen? I presume it had to do with boundary layer depth, but that should be explicit. Using fixed altitude ranges is convenient, but does it really reflect the structure of the atmosphere? It is often best to group data into mixed layer, cloud layer, buffer layer (if present), and free troposphere.

This information has been included in more detail in an overview of the ACE-ENA campaign submitted to the Bulletin of American Meteorological Society (Wang et al., 2021). Repeating it in this paper is out of its scope, as it is not intended to be a comprehensive review of the campaign.

Are there estimates of entrainment velocity during the experiment? That would constrain how fast the MBL is diluted with FT air and thus give a lower bound on MBL sources of aerosol. Or was the meteorological situation too complex for such approximations to make sense?

To our knowledge such estimates have not been made to date.

Since the literature survey was so minimal, I suppose it was inevitable that there would be a lack of direct comparison with earlier data from the region. But it would be interesting to know whether things have changed. Of particular interest is the fact that earlier experiments found much more $MS^-$. Is that a real change, or is it possibly that much of the $MS^-$ was on particles too large for the AMS? Were the $MS^-$:DMS or $MS^-$:$SO_4^{2-}$ ratios different? Is the aerosol sufficiently acidic to drive off $HNO_3$? (I strongly suspect that's why you saw so little $NO_3^-$; it was displaced to seasalt particles too big for the AMS to detect.)

The following paragraph was added, "The MSA to non-sea salt sulfate ratio (MSA:$SO_4$) measured during ACE-ENA in the summer was 0.02 on average in the MBL (<1000 m), which is lower than historical estimates of the ratio. For example, Pszenny et al. (1990) reports the ratio as 0.05 in the North Atlantic in August-September, Berresheim et al. (1991) reports 0.033 in Western North Atlantic in September, Savoie et al. (2002) reports 0.06 in Bermuda in September

and 0.05 in Mace Head in August. Huebert et al. (1996) found 0.07 in marine air masses and 0.02 in continental air masses in June in the Azores. The measurements reported in those earlier studies are based on analysis of filter samples, and they may not be directly comparable to AMS measurements reported here. In particular, because the filter samples can measure larger particles than the AMS, this might suggest that some of the MSA was present on coarse aerosols, such as sea salt."

I was also a bit surprised to see no estimates of how much of the aerosol in the boundary layer could have come from the free troposphere. Without level legs at the top of the MBL, you obviously cannot do a flux study, but you have tracers of long-distance transport that are immune to precipitation scavenging (CO, benzene, toluene) and at least 1 that is not (BC). Can you use those to put some bounds on how much of the $SO_4^{-2}$ and organic aerosol came from above? Would that bring the $MS^-$:$SO_4^{2-}$ in line with other work?

While this is a good suggestion for a follow-up study, accurate source apportionment is not straight-forward without chemical signatures that would correlate the aerosol with a source. Sulfate signatures in the AMS are generally the same for fresh and aged emissions, and while organics usually show some compositional diversity, a surprising observation in this campaign was uniformity of organic oxidation state (Figures 5 and 6). Adding a layer of complexity, it is expected that the structure of the MBL complicates simple application of back trajectories. Because of all these reasons, we found it appropriate not to speculate beyond the discussion in Figures 5 and 6.

The dramatic split in Fig. 6b in pHF is interesting. Since Table S1 says there were 5 profiles on 19 July, it seems likely that some profiles were in the putative fire plume while others were not. It might be worth trying multiple back trajectories on each of the profiles to establish a pattern like that seen in Clarke et al. (2013), where the ensemble of trajectories and their correlation with aerosol properties lent credibility to the trajectories over rather long distances.

Some of the ambiguity in the figure was removed by plotting only the vertical profile data in panels A and B. All of the vertical profiles happened to transect the plume. The figure in Clarke et al. (2013) looks very similar to Figure 14 in our paper. This is indeed what we were trying to do: generate multiple back-trajectories with systematic spatial offsets centered on one of the vertical profiles to make sure that we are not over-interpreting a single back-trajectory.

line 21 "fully" is meaningless. It suggests that all possibly relevant instrumentation was aboard, and that's not possible even on much bigger aircraft.

We removed "fully" from this sentence.

line 26 average submicrometer non-refractory aerosol mass

We added "sub-micrometer" to this sentence

line 33 "1 % of the sulfate and no more than 3 % of the total aerosol" makes no sense. You presumably meant no more than 3 % of the submicron organic aerosol.

This sentence was revised to read, "MSA accounted for no more than 3% of the sub-micron, non-refractory aerosol in the boundary layer."

line 140 Was there a typical altitude range of the spirals? "Through the atmosphere" is vague (and strictly speaking, incorrect). Was there a minimum altitude span criterion for inclusion in Table S1? Were there any criteria for where to do the spirals?

The spiral profiles were included in every flight to investigate the vertical structure. The altitude varied from flight to flight. Table S1 now includes the altitude information.

line 145 Was the standard aerodynamic lens used? What temperature was the vaporizer?

This information is now included, "The standard aerodynamic lens was used, and the AMS vaporizer was set to 600°C."

line 155 Has the inlet efficiency for this inlet been characterized?

A characterization of the inlet has not been published. Unpublished data from the G-1flight teams suggests the transmission efficiency is near 100% for particles smaller than 5 microns. The manufacturer's manual on the inlet states that the inlet "transmit particles with diameters between 0.01 and 6 micrometers with better than 95% efficiency." We added the following, ". The isokinetic inlet transmission efficiency is greater than 95% for 0.01 – 5 μm diameter particles."

line 156 "switched ... based on cloud cover" seems unlikely. Surely switching was done based on whether the plane was actually in cloud.

This was revised, "Sample streams between two inlets were switched by the instrument operator aboard the aircraft based on the presence of clouds."

lines 166–167 I'm a bit surprised at all the English (rather than SI) units. I guess that's up to the journal editors. In addition, the OD of the tubing is irrelevant; it's the ID that matters.

Stainless steel tubing is ¼" OD with .035" wall thickness. The Teflon is ¼" OD with 3/16" ID. The PEEK tubing is 1/16" OD with 0.055" ID. This was now re-stated, "The PTR-MS sampled air through a dedicated inlet that consisted of approximately 6" of 1/4" OD stainless steel with 0.035" wall thickness, followed by approximately 46" of 1/4" OD (3/16" ID) Teflon tubing, including a Teflon filter, and 36" of 1/16" OD (0.055" ID) PEEK tubing."

line 186 The low supersaturation is presented inconsistently. Here and in Fig. S3 it is 0.1 %, while in Fig. 13 it is 0.13 %.

It should be 0.13% everywhere. This was now fixed.

line 173 I don't see how elevated DMS background in the summer IOP necessarily biases DMS measurements high. If it is indeed an isobaric interference (any idea what would do that?) then one would eliminate the overestimate by subtracting the background. If it was incomplete destruction of the DMS, then subtracting the background would produce an underestimate. If the interfering species was partly destroyed by the catalyst, then yes you would have an overestimate of DMS, but you could only claim that the patterns you see in DMS are accurate if the interfering species concentration was fairly constant.

We were unable to get consistent background measurements for DMS; they were consistently higher at the beginning and end of flights relative to in-flight. We attribute this to two factors 1) adsorption of an unknown interfering species on tubing walls at the airport followed by slow desorption of this interfering species during flight and 2) incomplete removal of the interfering species by the catalyst. These values were clearly observed to decrease as the aircraft transited to cleaner air. We choose low DMS values observed in the FT as the background. As a result, DMS values measured in the presence of the interference were biased high. On several flights, porpoising maneuvers into and out of the boundary layer showed clear and sharp transitions in the DMS measurements. On all flights, clear changes in DMS concentrations were observed when transitioning into the FT. Therefore, we feel the relative changes in DMS concentrations are instructive while the absolute concentrations are not. We note that changes were made to the inlet lines that greatly mitigated this interference in the winter campaign.

line 182 "more closely mimic" is incomplete. More closely than what?

"more" has been removed.

line 229–230 Has anyone ever claimed that MSA accounted for the majority of particulate sulfate mass?

This sentence was misleading and was removed.

line 235 The equation is only true if those species are the only acids and bases present in the aerosol. There are organic acids like oxalic acid and MSA. They are probably negligible here, but that ought to be noted.

We added, "The analysis also neglects organic acids, such as oxalic acid or MSA, but the concentrations of these in clean MBL are low. MSA, for example, was shown to account for only 1-2% of the non-refractory aerosol mass."

lines 242–245 Be more explicit about the use of the thermal denuder. Was there very little submicron seasalt (as is likely)? As determined by heating to what temperature? It's true that the AMS doesn't see coarse particles, so whether they are externally mixed seems irrelevant. Is there a claim here that there was little volatile material on coarse particles so the AMS wasn't actually missing significant $MS^-$ and $NO_3^-$?

The thermodenuder was placed in front of the FIMS instrument, which measured the particles size distribution from 10 to 500 nm, and heated to 300°C to remove non-refractory material from

the particles. These measurements indicated there was little submicron sea-salt and were meant to indicate as such. We have deleted the reference to the sea-salt being externally mixed. The sentence now reads, "Use of thermodenuder heated to 300°C in front of an aerosol sizing instrument aboard the G-1 aircraft during the ACE-ENA campaign was used to infer that the AMS observations are of non-sea salt submicron marine aerosol."

line 250 It's not wrong, but a bit odd to use a moving average with an even number of points, meaning that the time represented by the average is between the times of two data points, but not right on them. I suppose it doesn't matter here since there aren't any comparisons here that depend on close synchronization.

It appears that Igor Pro defaults to an odd number of points when an even number of points is specified in the smoothing function. We changed the number of points in the text to 11.

line 275 Acid-catalyzed reactive uptake of organic vapors is an interesting idea. Any citations for it? Could SOA production via that mechanism be fast enough to account for the extra organic aerosol in the MBL?

Acid-catalyzed uptake of IEPOX has been shown to be more efficient at SOA production than non-IEPOX photochemical reactions. Some of the relevant literature includes Surratt et al. (2010), Lin et al. (2012) and D'Ambro et al. (2019). We added the following, "While the isoprene concentrations measured during ACE-ENA were low and close to the detection limit of 0.1 ppb (Table 2) SOA formation from acid-catalyzed IEPOX chemistry has been shown to be significantly more efficient than from non-IEPOX photochemical mechanisms (Surratt et al., 2010)."

line 371 (This is a hobbyhorse of mine.) Airmasses do not have origins! The air always came from somewhere earlier and has traces of that left in it. If there was near-total scavenging event or a large influx of pollution that dwarfs whatever was present, then one could claim there is an origin of the characteristics of the air mass. Ascribing an origin to an air parcel in the MBL is particularly absurd, since there is almost always entrainment mixing going on meaning that much of the air was in the FT within the last few days. Simple back trajectories are not really capable of conveying that.

This was rephrased, "Supplementary Figure S7 corroborates the low-altitude air masses as strongly influenced by marine conditions."

line 402 Surely you men "summer and winter" rather than "summer and fall".

Yes, this has been fixed.

Fig. 4 It's jarring that the pie charts for the organic fraction of the aerosol are larger than the the total aerosol.

We have resized the organic pie charts.

Fig. 6 This is an interesting figure, but the caption isn't as clear as it ought to be. It appears that in panels A and B, the organic loading is averaged into 100 m bins, while the pHF is for individual (10 s?) averages. The pHF data in panel B has a remarkable split between nearly neutralized and very acidic aerosol, as though some of the 5 spirals that day were in the pollution plume while the others were not. Of course the averaged organic loading cannot show that. Would it be worth plotting the spirals separately, or grouped as plume vs. non-plume? Were panels C, D, and E for the entire vertical profile? What does "normalized" mean here? Same area under the curve? It didn't happen often, but it appears from panel C that $MS^-$ was sometimes 10 % to 15 % of the $SO_4^{2-}$. Is that real?

The reviewer is right that there was ambiguity in panels A and B of this figure. The organic loadings were plotted only from vertical profiles, but the $pH_f$ values were plotted from the whole flight, including the level legs. This was fixed so that both the loadings and $pH_f$ values are from the vertical profiles. All the spirals sampled the plume, but the level legs did not.

Fig. 14 While the maps are quite pretty, there is a lot of information, such as bathymetry, that is unimportant to the paper. I don't actually object much to that even though it is best practice to exclude irrelevant material from graphics. However, including the political divisions within countries is clearly excessive.

We removed the political boundaries from the figure.

**Equations in the supplement** I assume the Copernicus editors will help you figure out what should be italicized and what should not be.

We will follow the editor's guidance on formatting.

**CE of MSA in the supplement** This is a misinterpretation of Middlebrook et al. (2012). It's not the pH that matters–it's whether the aerosol is liquid or solid. That said, since MSA salts are solid and MSA itself is liquid (much like H2SO4 and salts thereof), the CEs you propose are reasonable in the lab. I'm not sure what you're doing with the field data–the particles are presumably internal mixtures with only small contributions from $MS^-$. In that case, it's the presence of liquid $H_2SO_4$ that will determine CE for the entire aerosol.

This was re-stated as follows, "This is the largest source of uncertainty in translating the calibrations to ambient measurements of MSA, as ambient MSA is more acidic than the neutralized laboratory MSA, which implies that the ambient particles less viscous, which is expected to affect CE (Middlebrook et al., 2012)." It is indeed true that the presence of sulfuric acid determines the CE of ambient measurements and that's why we are using CE = 1 for all ambient data.

**PIKA vs Squirrel** Did you get ICH3SO2 + , ICH2SO2 + and ICH4SO3 + from PIKA or did you use unit mass data from Squirrel? I guess you did it with PIKA, which would make sense, particularly in the field, where other species would be present at those unit masses. To look for interferences, it might be useful to plot ICH4SO3 + vs ICH3SO2 + and ICH2SO2 + vs ICH4SO3 + to see whether you have the same fragmentation pattern in the field as you had in the lab.

We quantified these ions by fitting them in PIKA, as shown below. Evaluation of some of the individual HR fits suggests that there are no major interfering ions, especially in very clean conditions. In the field, these ions are of very low intensity, corresponding to low MSA concentrations, which makes comparisons to lab fragmentation patterns challenging. The plots below for June 28 and July 7 flight, when MSA was especially abundant suggest that the $CH_3SO_2/CH_2SO_2$ ratios are comparable in the lab and field (lab slope is ~3, field slope is ~2.5).

[Figure]

**Table S1** Including 4 digits after the decimal point for latitude and longitude specifies the point to within 11 m. Seems excessive. Might be useful to add an altitude range, unless that was constant (in which case I'd like to see that somewhere).

The latitude and longitude information was rounded up and altitude is now included.

**Fig. S3** The y axes on panels E and F are labeled SO4 rather than NH4.

The error was fixed.

**References**

[revised manuscript text omitted]